# Gaussian mixture layers for neural networks

**Sinho Chewi**                                                                        *sinho.chewi@yale.edu*
*Department of Statistics and Data Science*
*Yale University*

**Philippe Rigollet**                                                                     *rigollet@mit.edu*
*Department of Mathematics*
*Massachusetts Institute of Technology*

**Yuling Yan**                                                                          *yuling.yan@wisc.edu*
*Department of Statistics*
*University of Wisconsin–Madison*

**Reviewed on OpenReview:** *https://openreview.net/forum?id=sAptI2o5cP*

## Abstract

The mean-field theory for two-layer neural networks considers infinitely wide networks that are linearly parameterized by a probability measure over the parameter space. This nonparametric perspective has significantly advanced both the theoretical and conceptual understanding of neural networks, with substantial efforts made to validate its applicability to networks of moderate width. In this work, we explore the opposite direction, investigating whether dynamics can be directly implemented over probability measures. Specifically, we employ Gaussian mixture models as a flexible and expressive parametric family of distributions together with the theory of Wasserstein gradient flows to derive training dynamics for such measures. Our approach introduces a new type of layer—the Gaussian mixture (GM) layer—that can be integrated into neural network architectures. As a proof of concept, we validate our proposal through experiments on simple classification tasks, where a GM layer achieves test performance comparable to that of a two-layer fully connected network. Furthermore, we examine the behavior of these dynamics and demonstrate numerically that GM layers exhibit markedly different behavior compared to classical fully connected layers, even when the latter are large enough to be considered in the mean-field regime.

## 1 Introduction

Deep learning architectures are compositions of basic trainable *layers*, and many major milestones of deep learning models can be traced back to innovations for these fundamental components. For example, convolutional neural networks are based on *convolution layers* and *pooling layers*; residual neural networks are based on layers with *skip connections* (He et al., 2016); and more recently, the striking feats of large language models stem from the use of *attention layers* (Vaswani et al., 2017). Yet without guiding principles to map out the potential design space, the development of successful new layers has been largely elusive.

Whereas the vast majority of works on the theory of deep learning focus on understanding and explaining the behavior of existing architectures, in this paper we take the approach of applying the theory to propose new ones. In particular, we incorporate insights from the recent literature on the mean-field theory for neural networks (Chizat & Bach, 2018; Mei et al., 2018; 2019; Sirignano & Spiliopoulos, 2020; Chizat, 2022; Nitanda et al., 2022; Rotskoff & Vanden-Eijnden, 2022) in order to propose a new type of layer, which we call the *Gaussian mixture* (GM) *layer*.

According to mean-field theory, which we briefly review in Section 2, the training dynamics of a fully connected 2-layer neural network converges, under a scaling in which we divide the output by the number of neurons and

in the limit of infinite width, to a Wasserstein gradient flow in the space of probability measures. Here, the probability measure represents the *distribution* of weights in the fully connected layer and thereby shifts our perspective from the evolution of individual neurons to the evolution of the collection thereof. This framework has been successful at making nuanced and verifiable predictions about the training of certain simple neural networks (Mei et al., 2018; Abbe et al., 2023; Berthier et al., 2025), but thus far the mean-field limit has mostly been used as a tool for analysis. Indeed, it is not practical to actually implement the Wasserstein gradient flow, since doing so would require prohibitively large widths.

In this work, we explore the consequences of a *prescriptive* mean-field rather than a descriptive one. This proposal results in manipulating infinite dimensional objects—distributions over $\mathbb{R}^d$— and to implement it, we restrict the distribution over weights to a parametric *Gaussian mixture* whose parameters are trained via standard optimization routines. The use of mixture modelling can also be motivated on grounds of clustering phenomena for neurons but we do not explore this question here.

Although our GM layer is proposed as a replacement for a wide fully connected layer, the parametrization and restriction of the distribution of neurons to the class of finite Gaussian mixtures leads to markedly different training dynamics. We demonstrate this behavior through numerical experiments in Section 5 on the MNIST database, which also serve as a proof of concept for the incorporation of GM layers into neural network architecture design. Our initial results are promising and show that a GM layer attains comparable test performance to a 2-layer fully connected network. However, we stress that our goal is *not* to demonstrate superiority of the GM layer over existing architectures which would necessitate going beyond the simple two-layer architecture for which the mean-field theory applies. As a result, while we demonstrate that the GM layer is modular and can be integrated into deep architectures, we leave a detailed investigation to future research as it would require substantial engineering developments that are beyond the scope of this proposal.

**Contributions.**

- Our primary contribution is our Gaussian layer proposal, which we detail in Section 3.

- We also show that whereas the Wasserstein gradient flow over empirical measures is implemented via the Euclidean gradient flow over the locations, the gradient flow over Gaussian mixtures—equipped with a certain natural geometry—is implemented via the Euclidean gradient flow over the means and the *square roots* of the covariances (Theorem 1).

- In Section 4, we suggest an efficient parametrization scheme to speed up implementation.

- We conduct numerical experiments in Section 5 on the MNIST and Fashion-MNIST datasets, which serve as a proof of concept and provide some insights into the training dynamics of GM layers. First, we show that GM layers can achieve comparable performance as a 2-layer fully connected network. This is despite the fact that the two exhibit quite different training dynamics (see Figure 4). We also check that GM layers exhibit "feature learning"—as is expected in the mean-field regime—in the sense that the distribution over first layer weights moves substantially away from initialization. Finally, we exhibit performance gains obtained by going deeper (i.e., composing multiple GM layers).

**Other related work.**   In recent years, Wasserstein gradient flows have been applied to numerous probabilistic problems, such as sampling and variational inference. A common bottleneck for these applications is implementation of the flow, which can be achieved via stochastic dynamics in the case of sampling (Jordan et al., 1998) but remains challenging in general. The mean-field theory reviewed in Section 2 shows that the empirical distributions of the weights of a neural network along training indeed follow a Wasserstein gradient flow, but our goal in this work to maintain an evolution of a *continuous* distribution over the weights. The strategy we take here is to restrict the gradient flow to a parametric family, which was introduced in the context of variational inference for the family of Gaussians or mixtures of Gaussians (Lambert et al., 2022; Diao et al., 2023) and later applied to filtering (Lambert et al., 2023) and mean-field variational inference (Lacker, 2023; Yao & Yang, 2023; Ghosh et al., 2025; Jiang et al., 2025).

## 2 Review of mean-field theory

We briefly review the mean-field theory for 2-layer fully connected neural networks. A width-$m$ neural network computes a function $h_{\boldsymbol{\omega},\boldsymbol{\beta}} : \mathbb{R}^d \to \mathbb{R}$ of the form

$$h_{\boldsymbol{\omega},\boldsymbol{\beta}}(x) = \frac{1}{m} \sum_{j=1}^{m} \omega_j \ \diagup(\langle \beta_j, x \rangle),$$

where $\diagup : \mathbb{R} \to \mathbb{R}$ is the activation function—taken to be the ReLU function $\diagup(z) = \max(0, z)$ for the rest of this paper—and $(\omega_j, \beta_j) \in \mathbb{R} \times \mathbb{R}^d$ are trainable weights, for each neuron $j \in [m]$. Here and throughout, we use boldface to denote a collection of parameters. The $1/m$ scaling above is characteristic of the mean-field regime and enable the following perspective. Let $\rho^{(m)}$ denote the empirical distribution of the weights, $\rho^{(m)} = m^{-1} \sum_{j=1}^{m} \delta_{(\omega_j, \beta_j)}$,[1] then we can write $h_{\boldsymbol{\omega},\boldsymbol{\beta}}(x) = \int \omega \ \diagup(\langle \beta, x \rangle) \, \rho^{(m)}(\mathrm{d}\omega, \mathrm{d}\beta)$. In this formulation, however, we can make sense of this expression even when $\rho^{(m)}$ is no longer an empirical measure, and we can view $h$ as being parameterized by a probability measure $\rho$:

$$h_{\rho}(x) = \int \omega \ \diagup(\langle \beta, x \rangle) \, \rho(\mathrm{d}\omega, \mathrm{d}\beta). \tag{1}$$

Consider a loss[2] objective $\mathscr{L}$ and minimize the objective $\mathscr{L}(h_{\boldsymbol{\omega},\boldsymbol{\beta}})$ by following the gradient flow for the weights $\{\omega_j, \beta_j\}_{j \in [m]}$. This gives rise to a curve of parameters $(\boldsymbol{\omega}(t), \boldsymbol{\beta}(t))_{t \geq 0}$, and corresponding empirical measures $\rho^{(m)}(t) = m^{-1} \sum_{j=1}^{m} \delta_{(\omega_j(t), \beta_j(t))}$. The insight of mean-field theory is that the evolution of the empirical measures can be described as the (time-rescaled) gradient flow of the loss function $\rho \mapsto \mathscr{L}(h_{\rho})$ over the space of probability measures, equipped with the Wasserstein metric from optimal transport, initialized at $\rho^{(m)}(0)$. We refer to Villani (2003); Ambrosio et al. (2008); Villani (2009); Santambrogio (2015) for background on optimal transport and Wasserstein gradient flows.

The advantage of this reformulation is that it admits a well-defined limit as $m \to \infty$: if $\rho^{(m)}(0) \to \rho(0)$, then the curve of measures $(\rho^{(m)}(t))_{t \geq 0}$ converges to the Wasserstein gradient flow of $\rho \mapsto \mathscr{L}(h_{\rho})$, initialized at $\rho(0)$. This mean-field limit is, in some cases, easier to study than the original dynamics over the weights, and leads to predictions about the behavior of wide neural networks.

To summarize: the training dynamics of a finite-width neural network correspond to a Wasserstein gradient flow, initialized at (and remaining through its trajectory) an empirical measure, but the Wasserstein gradient flow picture is more general because it allows for flows of continuous measures. Unfortunately, in the latter case, the Wasserstein gradient flow does not readily lend itself to tractable implementation. The mean-field theory described above shows that it is *well-approximated* by a gradient flow started at an empirical measure, but this approximation often requires prohibitively large width. In the next section, we take the familiar approach from statistics of restricting the measures to a parametric family, namely, the set of finite Gaussian mixtures.

## 3 A mean-field theory over the space of Gaussian mixtures

We now introduce the Gaussian Mixture (GM) layer, beginning with the case of a single GM layer (corresponding with the mean-field theory described in Section 2). Here, we restrict the measure $\rho$ in (1) to be a Gaussian mixture with $K$ components:

$$\rho = \rho_{\boldsymbol{\mu},\boldsymbol{\Sigma}} := \frac{1}{K} \sum_{k=1}^{K} \mathcal{N}(\mu_k, \Sigma_k), \qquad \mu_k \in \mathbb{R}^{d+1}, \qquad \Sigma_k \in \mathbb{R}^{(d+1) \times (d+1)}. \tag{2}$$

Thus, the measure $\rho$ is now parameterized by a set of means and covariances for the components of the Gaussian mixture. We use the short-hand notation $h_{\boldsymbol{\mu},\boldsymbol{\Sigma}} := h_{\rho_{\boldsymbol{\mu},\boldsymbol{\Sigma}}}$. With this restriction, we can now train the

---

[1]Here, $\delta_x$ refers to a Dirac measure located at $x$.

[2]The loss typically depends on a training set, but this is irrelevant for our discussion here so it is omitted.

GM layer by minimizing $\mathscr{L}(h_{\boldsymbol{\mu},\boldsymbol{\Sigma}})$ with respect to the parameters $(\boldsymbol{\mu}, \boldsymbol{\Sigma})$. For example, in a regression task with a labeled dataset $\{x_i, y_i\}_{i \in [n]}$, we might take the squared loss defined by $\mathscr{L}(h) \coloneqq \sum_{i=1}^{n}(y_i - h(x_i))^2$.

The use of mixture modelling to model the distribution over neurons is motivated by the empirical observation that for many problems, the neurons tend to *cluster* (e.g., Papyan et al., 2020; Chen et al., 2023). Given this ansatz, the use of Gaussian mixtures emerges as a natural model for $\rho$, although other alternatives could be explored.

As discussed in Section 2, it is well-known from mean-field theory that the Wasserstein gradient flow restricted to empirical measures is implemented, up to time rescaling, by the Euclidean gradient flow with respect to the locations of the particles (Chewi et al., 2025, Proposition 6.16). When we move to Gaussian mixtures, we effectively replace the particles $\theta_j = (\omega_j, \beta_j)$ with "Gaussian particles" $\mathcal{N}(\mu_k, \Sigma_k)$, which are themselves distributions over $\theta$ but can be viewed simply as $(\mu_k, \Sigma_k)$ pairs. In the case $K = 1$ of a single Gaussian particle, it turns out that the Wasserstein gradient flow restricted to Gaussian measures is implemented simply by evolving the parameters $(\mu, C)$ via the (Euclidean) gradient flow, where $\Sigma = CC^\top$. This is usually known as the *Bures–Wasserstein gradient flow*, as discussed next.

## 3.1 Interpretation as a Wasserstein gradient flow

The gradient flows that we consider in this paper are closely related to Wasserstein gradient flows, as we describe in detail here.

**Bures–Wasserstein gradient flows.** We first consider the case $K = 1$, so that $\rho = \mathcal{N}(\mu, \Sigma)$ is simply a Gaussian measure. The space of non-degenerate Gaussian measures over $\mathbb{R}^D$, which is naturally identified with $\mathbb{R}^d \times \mathbf{S}_{++}^d$, can be equipped with the Wasserstein metric and is then known as the *Bures–Wasserstein space* (Bhatia et al., 2019). We denote this space by $\mathsf{BW}(\mathbb{R}^D)$. The Riemannian structure of the Wasserstein space endows $\mathsf{BW}(\mathbb{R}^D)$ with a Riemannian metric, called the *Bures–Wasserstein metric*.

Given a functional $\mathcal{L}$ over the Wasserstein space, we can restrict it to a functional $L : \mathsf{BW}(\mathbb{R}^D) \to \mathbb{R}$ via $L(\mu, \Sigma) \coloneqq \mathcal{L}(\mathcal{N}(\mu, \Sigma))$. If $\nabla_\mu$, $\nabla_\Sigma$ denote the usual Euclidean gradients of $L$ w.r.t. $\mu$ and $\Sigma$ respectively, it is known that the gradient flow of $L$ over $\mathsf{BW}(\mathbb{R}^D)$ is given by

$$\boxed{\dot{\mu} = -\nabla_\mu L(\mu, \Sigma)\,,} \qquad \text{and} \qquad \boxed{\dot{\Sigma} = -2\left(\Sigma\,\nabla_\Sigma L(\mu, \Sigma) + \nabla_\Sigma L(\mu, \Sigma)\,\Sigma\right).} \tag{3}$$

See, e.g., Altschuler et al. (2021, Appendix A) or Lambert et al. (2022, Appendix B.3). On the other hand, if we parameterize $\Sigma = UU^\top$ where $U \in \mathbb{R}^{D \times D}$ and we follow the Euclidean gradient flow of $(\mu, U) \mapsto L(\mu, UU^\top)$, it is straightforward to see that it coincides with (3). This parametrization also has the advantage of maintaining the positive semidefiniteness of $\Sigma$ along the optimization without the need for projections, which is convenient for implementation and can also be used to enforce low-rank factorizations (Burer & Monteiro, 2003). In Appendix A, we derive the Bures–Wasserstein gradient flows for our problems of interest, keeping in mind that they can also be implemented as Euclidean gradient flows via the parametrization $\Sigma = UU^\top$.

**Gaussian mixture gradient flows.** In the case $K \geq 2$, it is no longer possible to follow the Wasserstein gradient flow restricted to Gaussian mixtures. Indeed, the latter is not explicit, since the space of Gaussian mixtures is not a geodesically convex subset of the Wasserstein space. However, we can instead follow a Wasserstein gradient flow for the *mixing measure* $\nu \coloneqq \frac{1}{K} \sum_{k=1}^{K} \delta_{(\mu_k, \Sigma_k)}$ over the Bures–Wasserstein space. This geometric structure was previously introduced in Chen et al. (2019); Delon & Desolneux (2020) and can be interpreted as the Wasserstein geometry over the curved manifold $(\mathsf{BW}(\mathbb{R}^D), W_2)$, see Lambert et al. (2022, Appendix F) for details.

## 3.2 Gaussian mixture flow for neural networks

We prove that the Gaussian mixture flow admits a simple implementation (see Appendix A).

**Theorem 1.** *Let $\mathcal{L}$ be a loss function over the space of probability measures. Then, the Gaussian mixture gradient flow for $\mathcal{L}$ is equivalent, up to a time rescaling, to the Euclidean gradient flow of the objective $\mathcal{L}(h_{\boldsymbol{\mu},\boldsymbol{\Sigma}})$ with respect to the parameters $(\boldsymbol{\mu}, \boldsymbol{C})$, where $\Sigma_k = C_k C_k^\top$ for each $k \in [K]$.*

In other words, compared to ordinary neural networks, training GM layers is accomplished by simply incorporating gradient steps for the *square roots* of the covariances.

## 4 Implementation

In this section, we enhance the flexibility and tractability of GM layers by allowing for vector-valued outputs (needed for multi-class classification), reduced parametrization, and composition.

### 4.1 Multi-class classification and vector-valued outputs

Consider a multi-class classification problem with $L + 1$ labels denoted $\{0, \ldots, L\}$. Suppose we are given a dataset $\{x_i, y_i\}_{i \in [n]}$, where each $x_i \in \mathbb{R}^d$ and $y_i \in \{0, 1, \ldots, L\}$. It suffices to describe how to parameterize a vector-valued function $h : \mathbb{R}^d \to \mathbb{R}^L$, since we can then apply the logistic loss

$$\mathscr{L}(h) := -\sum_{i=1}^{n} \Big\{ \sum_{\ell=1}^{L} h(x_i)_\ell \mathbb{1}_{y_i = \ell} - \log\big(1 + \sum_{\ell=1}^{L} \exp(h(x_i)_\ell)\big) \Big\}.$$

The most straightforward way to parameterize the function $h : \mathbb{R}^d \to \mathbb{R}^L$ is via (1) and (2), where now $(\omega, \beta) \in \mathbb{R}^L \times \mathbb{R}^d$, i.e., the Gaussian mixture $\rho_{\boldsymbol{\mu}, \boldsymbol{\Sigma}}$ is a distribution over $\mathbb{R}^{d+L}$. In other words,

$$h_{\boldsymbol{\mu}, \boldsymbol{\Sigma}}(x) = K^{-1} \sum_{k \in [K]} \mathbb{E}_{(\omega, \beta) \sim \mathcal{N}(\mu_k, \Sigma_k)}[\omega \underline{\diagup}(\langle \beta, x \rangle)]. \tag{4}$$

However, the number of parameters becomes $\Theta((d + L)^2 K)$, which is prohibitively large.

### 4.2 Reducing the number of parameters

To reduce the number of parameters, we propose to incorporate sparsity into the model by considering only diagonal covariance matrices for $\beta$: $\beta \sim \mathcal{N}(\mu, \mathsf{diag}(\sigma^2))$ where $\mathsf{diag}(\sigma^2)$ is a diagonal matrix whose entries are given by the vector $\sigma^2 = \sigma \odot \sigma \in \mathbb{R}^d$.[3] Moreover, for jointly Gaussian $(\omega, \beta)$, the conditional mean of $\omega$ given $\beta$ is affine: $\mathbb{E}[\omega \mid \beta] = U\beta + v$. Since (4) only requires to know the conditional expectation $\mathbb{E}[\omega \mid \beta]$, we model each component of the Gaussian mixture as

$$\beta \sim \mathcal{N}(\mu, \mathsf{diag}(\sigma^2)), \qquad \mathbb{E}[\omega \mid \beta] = U\beta + v.$$

The trainable parameters for such a component are $\mu \in \mathbb{R}^d$, $\sigma \in \mathbb{R}^d$, $U \in \mathbb{R}^{L \times d}$, and $v \in \mathbb{R}^L$. Hence, the parameters for the full Gaussian mixture are $\boldsymbol{\theta} := (\boldsymbol{\mu}, \boldsymbol{\sigma}, \boldsymbol{U}, \boldsymbol{v}) = \{(\mu_k, \sigma_k, U_k, v_k)\}_{k \in [K]}$. This leads to

$$h_{\boldsymbol{\theta}}(x) = K^{-1} \sum_{k \in [K]} \mathbb{E}_{\beta \sim \mathcal{N}(\mu_k, \mathsf{diag}(\sigma_k^2))}[(U\beta + v) \underline{\diagup}(\langle \beta, x \rangle)]. \tag{5}$$

This use of (5) cuts down the number of parameters to $\Theta(dKL)$, a substantial savings when $L \ll d$ (as is typical). An additional benefit is that parametrization by $\boldsymbol{\sigma}$ automatically maintains the positive semidefiniteness of the covariances during training as they are of the form $\mathsf{diag}(\sigma^2)$, without recourse to costly projection steps.

To summarize, for multi-class classification, we train the parameters $\boldsymbol{\theta}$ via an optimization algorithm (in our experiments, we use vanilla stochastic gradient descent) on the objective $\mathscr{L}(h_{\boldsymbol{\theta}})$, where $\mathscr{L}$ is the multi-class logistic loss and $h_{\boldsymbol{\theta}}$ is given in (5).

**Comparison with fully connected networks.** In summary, a fully connected layer with input and output dimensions $d$ and $L$ respectively, and width $W$, has $\Theta((d + L) W)$ parameters. A GM layer with $K$ components and full covariance parametrization has $\Theta((d + L)^2 K)$ parameters. A GM layer with $K$ components and reduced parametrization has $\Theta(dKL)$ parameters. Typically, $K \ll W$, e.g., in our experiments we often take $K = 5$ or $K = 10$, and for a classification problem with not too many classes, $L \ll d$ as well. The parameter count directly translates into memory and computation time per gradient update.

---

[3]We may also write $\mu = \mu^\beta$ to avoid confusion with (2).

### 4.3 Stacking GM layers

In the previous subsection, we showed how to parameterize a function $h : \mathbb{R}^d \to \mathbb{R}^L$ as a GM layer. Since the input and output dimensions are arbitrary, this construction can be readily composed with other types of layers—including GM layers themselves—in order to build up deep neural network architectures. Indeed, as depicted in Figure 1, the GM layer can be dropped in as a replacement for an (infinitely wide) fully connected layer and, so long as a square root parameterization is used for the covariance, can be trained via standard backpropagation by Theorem 1.

We leave the question of designing and optimizing deep architectures that integrate the GM layer to future research. Instead, we focus on a single GM layer, and call it a *GM network*.

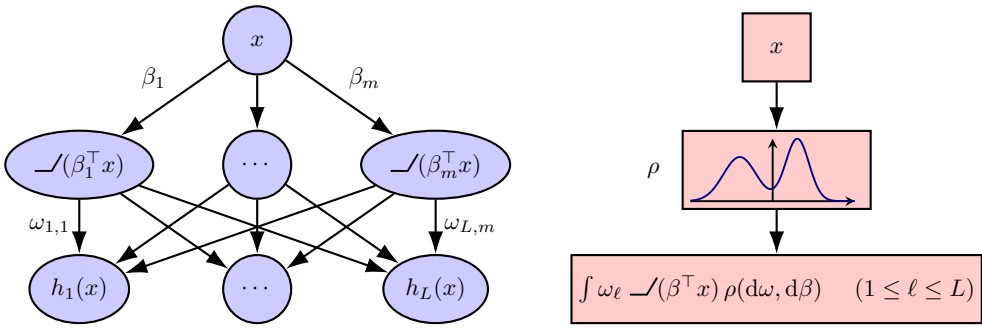

Figure 1: A GM layer (right) can act as a replacement for a fully connected layer (left).

## 5 Experiments

The source code for all experiments in this section is available at https://github.com/yulingy/GM_layer.

**Dataset.** We test the performance of neural networks with GM layers on multi-class classification on two widely used datasets: MNIST (LeCun & Cortes, 2010) and Fashion-MNIST (Xiao et al., 2017). Both datasets consist of 60,000 training examples and 10,000 test examples, where each example is a $28 \times 28$ grayscale image, associated with a label from one of 10 classes. Each image is vectorized and normalized to have zero mean and unit standard deviation. Throughout this section, test error refers to the misclassification error evaluated over the test set.

**Setup.** The number of components $K$ is a hyperparameter: larger $K$ enables more expressive GM layers, while smaller $K$ speeds up computation. We consider $K \in \{5, 10, 20\}$ for different experiments. For a GM layer with parameters $\mu^\beta$, $\sigma$, $U$, and $v$, we initialize the entries of $\mu^\beta$, $U$ and $v$ i.i.d. from $\mathcal{N}(0, \gamma^2)$, and the entries of $\sigma$ all equal to $\gamma$, for some $\gamma > 0$. For most of the experiments we fix $\gamma = 1/2$, but by adjusting the value of $\gamma$ we can investigate the role of the initialization scale, as discussed below. We train the network using SGD with batch size 64 and fixed learning rate 1 for the parameter $\sigma$ and 0.1 for all other parameters. All of these choices are used for simplicity and were found via minimal tuning.

**Test error.** For both datasets, we test the performance of a network with one GM layer, where the number of components $K$ takes values in $\{5, 10, 20\}$. The results are presented in Figure 2. As we can see, a single GM layer with 20 components achieves a test error of $\approx 2.77\%$ for MNIST, and $\approx 12.13\%$ for Fashion-MNIST. Increasing $K$ leads to better performance, but the marginal improvement is very small after $K$ exceeds 10, suggesting that $K = 10$ already leads to sufficiently expressive Gaussian mixtures. In practice, one may choose $K$ between 10 and 20 to strike a balance between expressive power and computational efficiency.

We also compare the test error of the GM network with the 2-layer fully connected network. We set the width of the latter network $m$ to be 1000 (in fact the test error curve is almost the same for $m \in \{100, 500, 1000\}$, so we just present the result for $m = 1000$ for simplicity). We use two different methods to initialize the

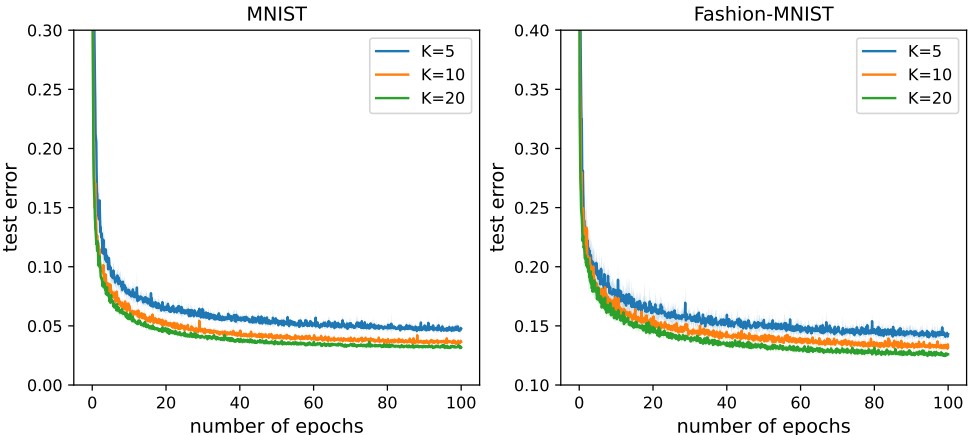

Figure 2: Test error (with error bars) for a GM Network with $K = 5, 10, 20$ number of components vs. the number of epochs. The left (resp. right) panel shows the result for MNIST (resp. Fashion-MNIST) dataset. The error bars are computed over 5 independent trials.

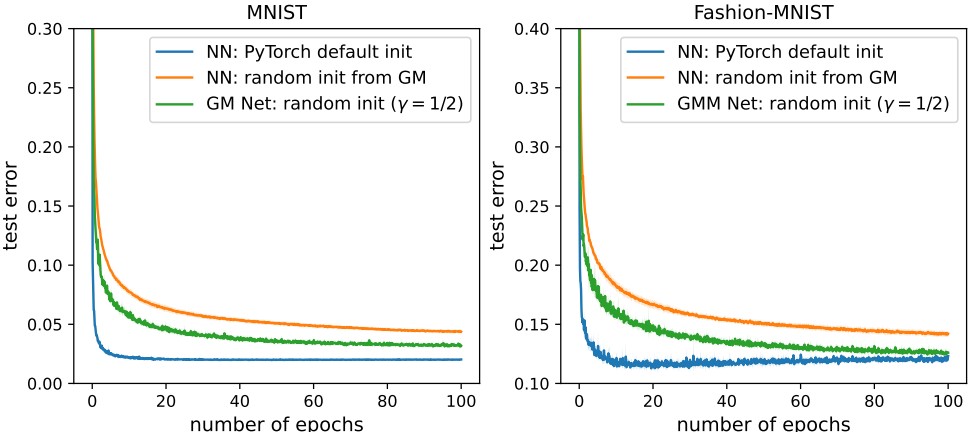

Figure 3: Test error (with error bars) for a 2-layer fully connected network using two different initialization schemes vs. the number of epochs. The left (resp. right) panel shows the results for the MNIST (resp. Fashion-MNIST) dataset. The error bars are computed over 5 independent trials.

parameters of a fully-connected layer: PyTorch's default method (Kaiming uniform (He et al., 2015)), and random initialization drawn i.i.d from the initial distribution we used in training the GM network with $K = 20$ components (for a fair comparison with the GM network). The results are presented in Figure 3. We can see that although in the end all curves converges to comparable test errors, the number of epochs required to achieve low test are different: the GM network performs better than fully connected network with matching initialization, i.e., with weights drawn randomly from the initial Gaussian mixture distribution of the GM network, but worse than that with PyTorch's default initialization. This observation raises the question of whether the slight under-performance of the GM network compared to the fully connected network with default initialization could be overcome via better initialization schemes for GM layers.

**Evolution of Gaussian components.** In order to visualize the evolution of the Gaussian mixture $(\rho_t)_{t \geq 0}$ during the training phase, we train the network for $T = 200$ epochs, compute the top 2 principal subspace of the final Gaussian mixture distribution $\rho_T$ (marginalized over $\beta$), and project the entire training trajectory $(\rho_t)_{0 \leq t \leq T}$ (marginalized over $\beta$) onto this 2-dimensional subspace. Figure 4 depicts the evolution of this 2-dimensional distribution across 200 epochs for $K = 5$, using two types of initialization schemes $\gamma \in \{1/2, 1/256\}$. As we can see, the means of the five Gaussian components move far away from their

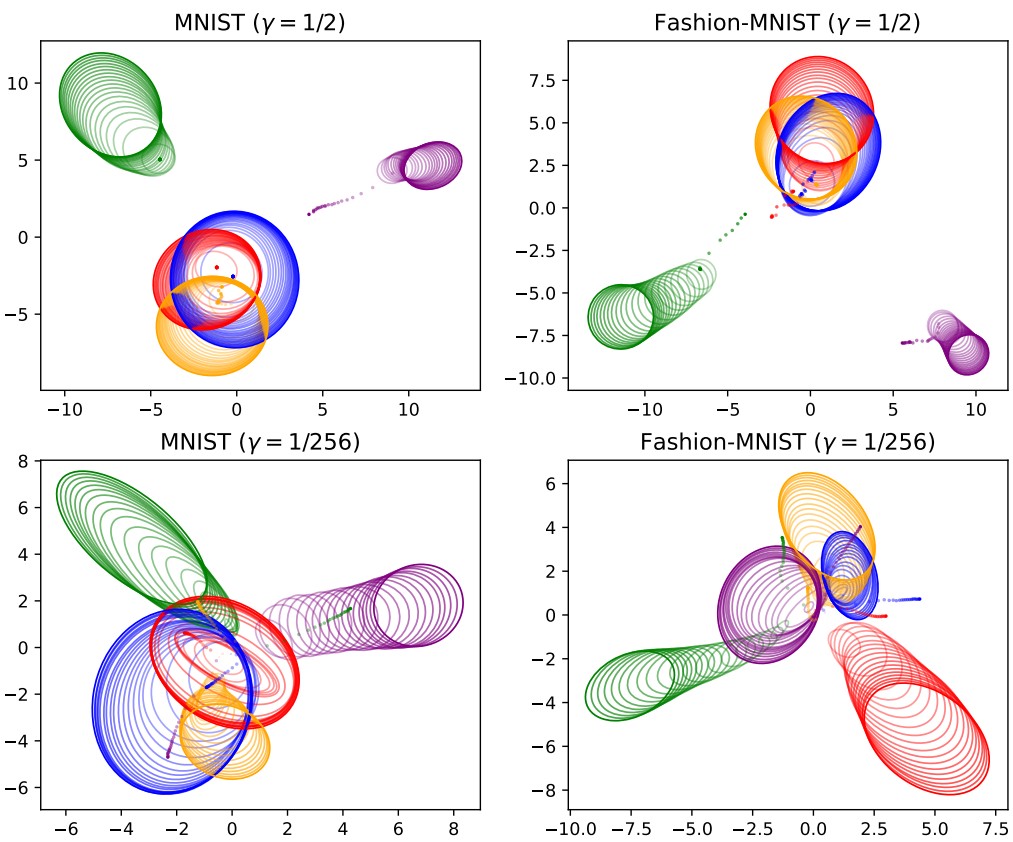

Figure 4: The evolution of the Gaussian components (marginalized over $\beta$) and weights $\beta_j$ of the neurons, projected onto the top 2 PCs of the final GM distribution, across 200 epochs of training. The number of components for the GM net (resp. number of neurons in the 2-layer neural network) is $K = 5$. The projected Gaussian components are represented by their covariance ellipses centered at their means, while the projected weights of the neurons are depicted as dots. We use the same color for the evolution of the same Gaussian component and neuron, with increasing opacity as the number of epochs increases. The left (resp. right) plots show results for MNIST (resp. Fashion-MNIST), while the top (resp. bottom) plots use initialization scale $\gamma = 1/2$ (resp. $\gamma = 1/256$).

initializations (even when they are initialized near zero, which is the case when $\gamma = 1/256$), and the covariance matrices also become non-isotropic quickly. This also shows that the training dynamics of networks with GM layers are not overly sensitive to initialization.

We also train a fully-connected 2-layer neural network with width $m = K$ to compare the training dynamics of networks with a GM layer and a fully-connected layer. To make the comparison fair, we initialize the neurons at $\beta_k = \mu_k$ and $\omega_k = U\beta_k + v$ for $1 \leq k \leq K$ where $\{\mu_k\}_{1 \leq k \leq K}$, $U$ and $v$ are the parameters of the initial Gaussian mixture distribution. We train the neural network using the same SGD algorithm with learning rate 0.1. The evolution of these neurons (projected onto the same 2-dimensional subspace) across 200 epochs is also shown in Figure 4. We can see that they exhibit drastically different training dynamics from that of the GM network. For example, when initialized at scale $\gamma = 1/2$, the Gaussian components of the GM layer tend to move far away from zero, while the neurons of the fully-connected layer do not exhibit this trend.

**Mean field vs. "NTK" regime.** We design a simple experiment to inspect whether the training of a network with a single GM layer is in the neural tangent kernel (NTK) a.k.a "lazy training" regime (Jacot et al., 2018; Chizat et al., 2019; Du et al., 2019; Bartlett et al., 2021), or the mean-field regime. Indeed, since the presence of feature learning is a powerful motivation for the mean-field regime, it is important to check

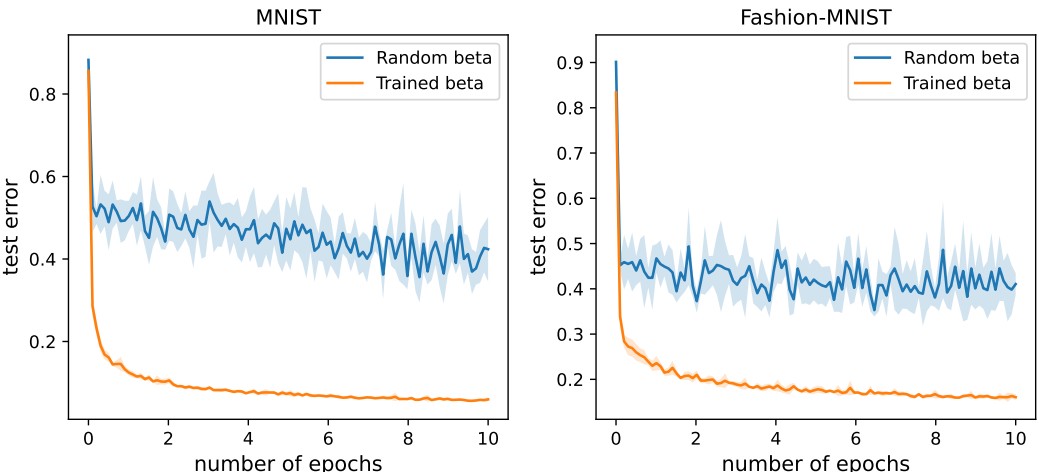

Figure 5: Test error (with error bars) for training with or without updating the marginal distribution over $\beta$ vs. the number of epochs. The left (resp. right) panel shows the result for the MNIST (resp. Fashion-MNIST) dataset. The error bars are computed over 5 independent trials.

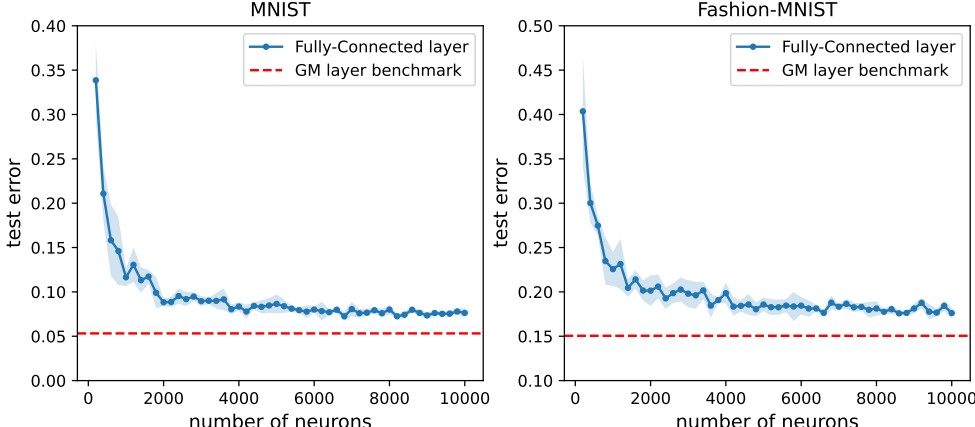

Figure 6: Test error (with error bars) for fully connected 2-layer networks constructed via subsampling vs. width. The neurons are sampled from a trained GM layer, whose test error is also plotted as a benchmark. The left (resp. right) panel shows the result for the MNIST (resp. Fashion-MNIST) dataset. The error bars are computed over 5 independent trials.

that this intuition also carries over to GM layers. Although lazy training is not formally defined for GM layers, we can loosely take it to be the case when the distribution over the "first layer weights" $\beta$ does not significantly move away from its initialization.

We fix the marginal distribution over $\beta$ at its initialization (which can be achieved by setting the learning rates for $\mu^\beta$ and $\sigma$ to be 0) and only update $U$ and $v$ for each Gaussian component. Figure 5 compares the performance of training the network with fixed $\beta$ vs. the network with trained $\beta$. If we fix the marginal distribution over $\beta$, the trained network can only achieve a test error of $\approx 40\%$ for both the MNIST and Fashion-MNIST datasets. This shows that good test performance can only be achieved if $\beta$ moves significantly from its initialization, and thus "feature learning" is indeed crucial for the performance of GM layers.

**Monte Carlo reduction to fully connected layers.** After training a network with a GM layer, which gives a distribution jointly over $(\omega, \beta)$, one natural question is whether it is possible to construct a fully connected 2-layer network with similar performance by sampling a reasonable number of neurons from this

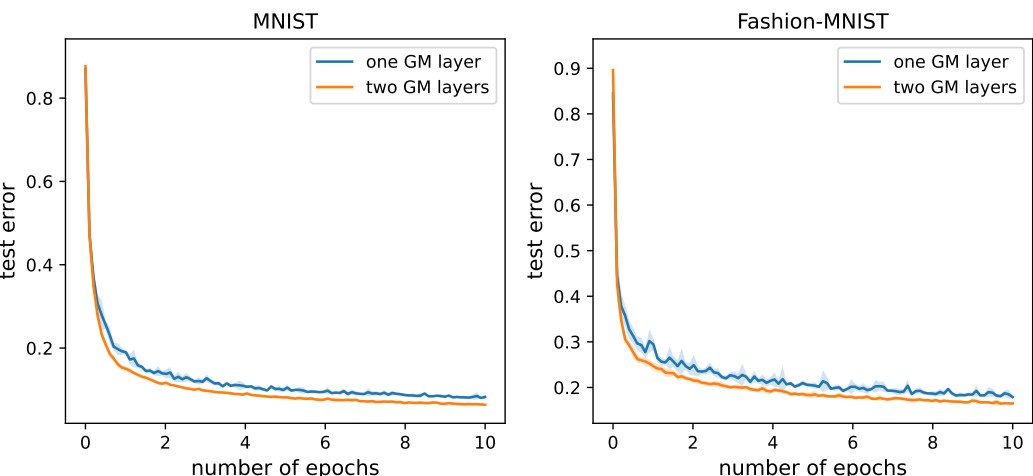

Figure 7: Test error (with error bars) for networks with one or two GM layers vs. the number of epochs. The left (resp. right) panel shows the result for the MNIST (resp. Fashion-MNIST) dataset. The error bars are computed over 5 independent trials.

Gaussian mixture distribution. To answer this question, we first train a network with a GM layer with $K = 20$ components and 20 epochs on MNIST. Then we construct fully connected 2-layer networks by sampling $m$ neurons (i.e., $(\omega, \beta)$ pairs) from the trained Gaussian mixture distribution, and evaluate their test errors without training. The results reported in in Figure 6 are indicating of the classical $1/\sqrt{m}$ convergence rate of Monte Carlo approximation. Unfortunately, this convergence is to slow from the perspective of test error: even for $m = 10^4$ there is still a gap between the performance of the GM network and its Monte Carlo approximation. This is a consequence of the high-dimensional nature of the space of parameter: we sample, on average $10^4/20 = 500$ points per component, and this is not sufficient to estimate accurately a Gaussian integral in dimension $28 \times 28 = 784$.

**Stacking multiple GM layers.** In the previous experiments, we used networks with a single GM layer. In this experiment, we demonstrate that stacking GM layers can improve classification performance. Specifically, we stack a GM layer with an input dimension of 784 and an output dimension of $L' = 100$ with a second GM layer with an input dimension of $L' = 100$ and an output dimension of $L = 9$, both layers utilizing $K = 10$ Gaussian components. This configuration yields a lower test error compared to a single GM layer with the same number of Gaussian components, as shown in Figure 7. Our primary objective is to provide a straightforward example illustrating the benefits of stacking multiple GM layers. We have not optimized hyperparameters such as the number of components $K$ and the width of the middle layer $m$ for optimal performance. Additionally, constructing a deeper architecture to achieve competitive performance on classic benchmark tasks like image classification would necessitate the integration and optimization of convolutional layers. This would shift the focus to secondary aspects that are beyond the scope of this paper.

**Experiments on CIFAR datasets.** Finally, we conduct an experiment based on the CIFAR-10 and CIFAR-100 datasets (Krizhevsky et al., 2009), which are larger in scale. We start with an effective VGG-like network architecture, which consists of four convolutional layers with batch normalization and max pooling, followed by three fully connected layers with dropout (see Appendix B for details). This network structure achieves a test error 12.62% for CIFAR-10, and a test error 61.48% for CIFAR-100 over 50 training epochs. To incorporate the GM layer into this structure, we replace the last two fully connected layers with a GM layer (with $K = 10$ components), which leads to an improved test errors 11.04% for CIFAR-10 and 43.55% for CIFAR-100, as shown in Figure 8. In addition to demonstrating the applicability of GM layers on more challenging datasets, this experiment also shows that GM layers can be composed with other types of layers and thus incorporated into complex architectures.

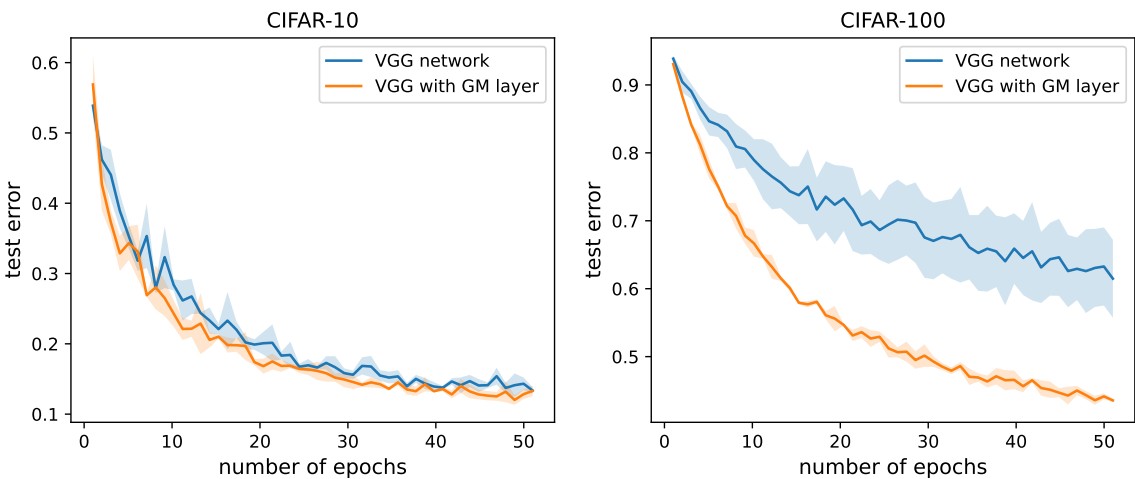

Figure 8: Test error (with error bars) for classifying CIFAR-10 and CIFAR-100 datasets using VGG-type network (with and without GM layers) vs. the number of epochs. The error bars are computed over 3 independent trials.

## 6 Conclusion

In this paper, we have introduced GM layers as a novel layer type for neural network architectures, offering a fresh perspective that bridges concepts from mean-field theory and variational inference. This approach opens up a wealth of unexplored possibilities for layer design, suggesting new avenues for incorporating diverse "variational families" to model $\rho$ beyond Gaussian mixtures, and for optimizing with respect to various geometries within the variational family. For instance, while our current model restricts to Gaussian mixtures with *equal weights*, this limitation can be addressed by optimizing over Gaussian mixtures with unequal weights via the Wasserstein–Fisher–Rao geometry (Liero et al., 2016; Chizat et al., 2018; Liero et al., 2018; Lu et al., 2019; Yan et al., 2024), as demonstrated in Lambert et al. (2022, Appendix H). Such advancements may also inspire alternatives for other types of layers, including convolutional and attention layers.

A notable limitation of our work is the preliminary nature of our experiments. Further empirical investigation is required to refine our design choices and compare them with existing architectures. For example, the sparse parametrization in Subsection 4.2 is quite drastic, suggesting the need to explore alternatives like low-rank factorization of the covariance matrix. Moreover, developing effective initialization and training strategies for GM layers remains an open question. We anticipate that addressing these challenges will pave the way for more robust and versatile neural network architectures in the future.

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

## A   Exact derivations

In this section, we record exact expressions for the Gaussian mixture gradient flows. These expressions were used to validate the correctness of our GM layer implementations in `PyTorch` and could be useful for future investigations.

We consider the following two problems.

- **Regression.** We are given a dataset $\{(x_i, y_i)\}_{i \in [n]}$ where each $x_i \in \mathbb{R}^d$ and $y_i \in \mathbb{R}$. We consider the squared loss

$$\mathscr{L}(h) = \frac{1}{n} \sum_{i=1}^{n} \left(y_i - h(x_i)\right)^2.$$

- **Multi-class classification.** We are given a dataset $\{(x_i, y_i)\}_{i \in [n]}$, where each $x_i \in \mathbb{R}^d$ and $y_i \in \{0, 1, \ldots, L\}$ and $L$ denotes the number of classes. For this problem, we consider the multi-class logistic loss

$$\mathscr{L}(h) = -\frac{1}{n} \sum_{i=1}^{n} \left\{ \sum_{\ell=1}^{L} h(x_i)_\ell \mathbb{1}_{y_i=\ell} - \log\left(1 + \sum_{\ell=1}^{L} \exp\left(h(x_i)_\ell\right)\right) \right\},$$

  where $h : \mathbb{R}^d \to \mathbb{R}^L$.

For regression, we derive exact expressions for the gradient flows for the case of $K = 1$ (i.e., the Gaussian mixture is simply a Gaussian) and for the case $K > 1$ for the full parametrization (4). For multi-class classification, we focus on the case $K > 1$ with the diagonal parametrization (5).

**Notation.** For $\theta = (\omega, \beta)$, we use the shorthand notation $f(x, \theta) := \omega \, \lrcorner(\langle \beta, x \rangle)$. We write $\phi(\cdot \mid \mu, \Sigma)$ for the density of $\mathcal{N}(\mu, \Sigma)$. We also use $\mathcal{L}$ to denote a loss over the space of probability measures (whereas $\mathscr{L}$ denotes a loss over the space of functions).

### A.1   Regression

#### A.1.1   Gaussians

We first restrict $\rho$ to be a single Gaussian, namely

$$\mathbb{R}^D \ni \theta = \begin{bmatrix} \omega \\ \beta \end{bmatrix} \sim \rho = \mathcal{N}(\mu, \Sigma) = \mathcal{N}\left( \begin{bmatrix} \mu^\omega \\ \mu^\beta \end{bmatrix}, \begin{bmatrix} \Sigma^\omega & \Sigma^{\omega,\beta} \\ \Sigma^{\beta,\omega} & \Sigma^\beta \end{bmatrix} \right),$$

where $D = d + 1$. Then we can write the objective function as

$$\min_{\rho = \mathcal{N}(\mu, \Sigma)} \mathcal{L}(\rho) := \frac{1}{n} \sum_{i=1}^{n} \left( y_i - \int f(x_i, \theta) \, \rho(\mathrm{d}\theta) \right)^2. \tag{6}$$

The Bures–Wasserstein gradient flow for minimizing $\mathcal{L}$ is characterized by the following theorem.

**Theorem 2.** *The Bures–Wasserstein gradient flow $(\rho_t)_{t \geq 0}$ for minimizing $\mathcal{L}$ in (6) is given by $\rho_t = \mathcal{N}(\mu_t, \Sigma_t)$ that evolves according to the ODE system*

$$
\boxed{
\begin{aligned}
\dot{\mu}_t &= \frac{2}{n} \sum_{i=1}^{n} \big(y_i - \mathbb{E}_{\rho_t} f(x_i, \theta)\big) \, \mathbb{E}_{\rho_t} \nabla_\theta f(x_i, \theta) \,, \\
\dot{\Sigma}_t &= \frac{2}{n} \sum_{i=1}^{n} \big(y_i - \mathbb{E}_{\rho_t} f(x_i, \theta)\big) \, \mathbb{E}_{\rho_t} [\nabla_\theta f(x_i, \theta) \otimes (\theta - \mu_t) + (\theta - \mu_t) \otimes \nabla_\theta f(x_i, \theta)] \,.
\end{aligned}
}
$$

*Proof.* The loss function $\mathcal{L}$ for the regression problem (6) can be written as

$$
\begin{aligned}
\mathcal{L}(\rho) &= \frac{1}{n} \sum_{i=1}^{n} \left( y_i - \int f(x_i, \theta) \, \rho(\mathrm{d}\theta) \right)^2 \\
&= \frac{1}{n} \sum_{i=1}^{n} y_i^2 - \frac{2}{n} \sum_{i=1}^{n} y_i \int f(x_i, \theta) \, \rho(\mathrm{d}\theta) + \frac{1}{n} \sum_{i=1}^{n} \left( \int f(x_i, \theta) \, \rho(\mathrm{d}\theta) \right)^2 \\
&= \ell_0 + 2 \int V(\theta) \, \rho(\mathrm{d}\theta) + \iint U(\theta, \theta') \, \rho(\mathrm{d}\theta) \, \rho(\mathrm{d}\theta') \,,
\end{aligned}
$$

where $\ell_0 = n^{-1} \sum_{i=1}^{n} y_i^2$ is some constant that does not depend on $\rho$, and the two functions $V : \mathbb{R}^D \to \mathbb{R}$ and $U : \mathbb{R}^D \times \mathbb{R}^D \to \mathbb{R}$ are given by

$$
V(\theta) := -\frac{1}{n} \sum_{i=1}^{n} y_i f(x_i, \theta) \qquad \text{and} \qquad U(\theta, \theta') := \frac{1}{n} \sum_{i=1}^{n} f(x_i, \theta) \, f(x_i, \theta') \,. \tag{7}
$$

By (3), it suffices to compute the Euclidean gradients w.r.t. the variables $\mu$ and $\Sigma$. We first compute $\nabla_\mu \mathcal{L}(\rho)$ as follows:

$$
\begin{aligned}
\nabla_\mu \mathcal{L}(\rho) &= 2 \, \nabla_\mu \int V(\theta) \, \phi(\theta \mid \mu, \Sigma) \, \mathrm{d}\theta + \nabla_\mu \iint U(\theta, \theta') \, \phi(\theta \mid \mu, \Sigma) \, \phi(\theta' \mid \mu, \Sigma) \, \mathrm{d}\theta \, \mathrm{d}\theta' \\
&\overset{\text{(i)}}{=} 2 \int V(\theta) \, \nabla_\mu \phi(\theta \mid \mu, \Sigma) \, \mathrm{d}\theta + \iint U(\theta, \theta') \, \nabla_\mu [\phi(\theta \mid \mu, \Sigma) \, \phi(\theta' \mid \mu, \Sigma)] \, \mathrm{d}\theta \, \mathrm{d}\theta' \\
&\overset{\text{(ii)}}{=} 2 \int V(\theta) \, \nabla_\mu \phi(\theta \mid \mu, \Sigma) \, \mathrm{d}\theta + 2 \int \left[ \int U(\theta, \theta') \, \nabla_\mu \phi(\theta \mid \mu, \Sigma) \, \mathrm{d}\theta \right] \phi(\theta' \mid \mu, \Sigma) \, \mathrm{d}\theta' \\
&\overset{\text{(iii)}}{=} -2 \int V(\theta) \, \nabla_\theta \phi(\theta \mid \mu, \Sigma) \, \mathrm{d}\theta - 2 \int \left[ \int U(\theta, \theta') \, \nabla_\theta \phi(\theta \mid \mu, \Sigma) \, \mathrm{d}\theta \right] \phi(\theta' \mid \mu, \Sigma) \, \mathrm{d}\theta' \\
&\overset{\text{(iv)}}{=} 2 \int \nabla_\theta V(\theta) \, \phi(\theta \mid \mu, \Sigma) \, \mathrm{d}\theta + 2 \int \left[ \int \nabla_\theta U(\theta, \theta') \, \phi(\theta \mid \mu, \Sigma) \, \mathrm{d}\theta \right] \phi(\theta' \mid \mu, \Sigma) \, \mathrm{d}\theta' \\
&= 2 \int \left[ \nabla_\theta V(\theta) + \int \nabla_\theta U(\theta, \theta') \, \phi(\theta' \mid \mu, \Sigma) \, \mathrm{d}\theta' \right] \phi(\theta \mid \mu, \Sigma) \, \mathrm{d}\theta \tag{8}
\end{aligned}
$$

Here, step (i) uses the Leibniz integral rule (since $U$ and $V$ are continuous and $\phi$ is sufficiently smooth); step (ii) follows from the chain rule and the fact that $U(\theta, \theta') = U(\theta', \theta)$; step (iii) follows from (15) in Appendix A.3; and step (iv) follows from integration by parts. Following similar steps, we can compute

$\nabla_\Sigma \mathcal{L}(\rho)$ as follows:

$$
\begin{aligned}
\nabla_\Sigma \mathcal{L}(\rho) &= 2 \int V(\theta) \, \nabla_\Sigma \phi(\theta \mid \mu, \Sigma) \, \mathrm{d}\theta + 2 \int \left[ \int U(\theta, \theta') \, \nabla_\Sigma \phi(\theta \mid \mu, \Sigma) \, \mathrm{d}\theta \right] \phi(\theta' \mid \mu, \Sigma) \, \mathrm{d}\theta' \\
&\overset{(a)}{=} \int V(\theta) \, \nabla_\theta^2 \phi(\theta \mid \mu, \Sigma) \, \mathrm{d}\theta + \int \left[ \int U(\theta, \theta') \, \nabla_\theta^2 \, \phi(\theta \mid \mu, \Sigma) \, \mathrm{d}\theta \right] \phi(\theta' \mid \mu, \Sigma) \, \mathrm{d}\theta' \\
&\overset{(b)}{=} -\int \nabla_\theta V(\theta) \otimes \nabla_\theta \phi(\theta \mid \mu, \Sigma) \, \mathrm{d}\theta \\
&\quad\; -\int \left[ \int \nabla_\theta U(\theta, \theta') \otimes \nabla_\theta \phi(\theta \mid \mu, \Sigma) \, \mathrm{d}\theta \right] \phi(\theta' \mid \mu, \Sigma) \, \mathrm{d}\theta' \\
&\overset{(c)}{=} \int \nabla_\theta V(\theta) \otimes (\theta - \mu) \, \phi(\theta; \mu, \Sigma) \, \mathrm{d}\theta \, \Sigma^{-1} \\
&\quad\; +\int \left[ \int \nabla_\theta U(\theta, \theta') \otimes (\theta - \mu) \, \phi(\theta; \mu, \Sigma) \, \mathrm{d}\theta \right] \phi(\theta' \mid \mu, \Sigma) \, \mathrm{d}\theta' \, \Sigma^{-1} \\
&\overset{(b')}{=} -\int \nabla_\theta \phi(\theta \mid \mu, \Sigma) \otimes \nabla_\theta V(\theta) \, \mathrm{d}\theta \\
&\quad\; -\int \left[ \int \nabla_\theta \phi(\theta \mid \mu, \Sigma) \otimes \nabla_\theta U(\theta, \theta') \, \mathrm{d}\theta \right] \phi(\theta' \mid \mu, \Sigma) \, \mathrm{d}\theta' \\
&\overset{(c')}{=} \Sigma^{-1} \int (\theta - \mu) \otimes \nabla_\theta V(\theta) \, \phi(\theta; \mu, \Sigma) \, \mathrm{d}\theta \\
&\quad\; + \Sigma^{-1} \int \left[ \int (\theta - \mu) \otimes \nabla_\theta U(\theta, \theta') \, \phi(\theta; \mu, \Sigma) \, \mathrm{d}\theta \right] \phi(\theta' \mid \mu, \Sigma) \, \mathrm{d}\theta'
\end{aligned}
$$

Here, step (a) follows from (16) in Appendix A.3; steps (b) and (b') both follow from applying integration by parts to the formula following step (a); whereas steps (c) and (c') both follow from (14) in Appendix A.3. According to (3), the Bures–Wasserstein gradient flow for minimizing $\mathcal{L}$ is given by $\rho_t = \mathcal{N}(\mu_t, \Sigma_t)$, $t \geq 0$, where

$$
\begin{aligned}
\dot{\mu}_t &= -\nabla_\mu \mathcal{L}(\rho_t), \\
\dot{\Sigma}_t &= -2 \, \nabla_\Sigma \mathcal{L}(\rho_t) \, \Sigma_t - 2 \, \Sigma_t \, \nabla_\Sigma \mathcal{L}(\rho_t) \, .
\end{aligned}
$$

In order to derive a closed-form expression, we compute

$$
\nabla_\theta V(\theta) = -\frac{1}{n} \sum_{i=1}^n y_i \, \nabla_\theta f(x_i, \theta) = -\frac{1}{n} \sum_{i=1}^n y_i \begin{bmatrix} \diagup(\beta^\top x_i) \\ \omega \diagup'(\beta^\top x_i) \, x_i \end{bmatrix}, \qquad \text{and} \tag{9}
$$

$$
\nabla_\theta U(\theta, \theta') = \frac{1}{n} \sum_{i=1}^n \nabla_\theta f(x_i, \theta) \, f(x_i, \theta') = \frac{1}{n} \sum_{i=1}^n f(x_i, \theta') \begin{bmatrix} \diagup(\beta^\top x_i) \\ \omega \diagup'(\beta^\top x_i) \, x_i \end{bmatrix}. \tag{10}
$$

where $\diagup'(x) = \mathbb{1}\{x \geq 0\}$ is the derivative of ReLU at any $x \neq 0$.[4] Hence, we arrive at

$$
\begin{aligned}
\dot{\mu}_t = -\nabla_\mu \mathcal{L}(\rho_t) &= -2 \int \left[ \nabla_\theta V(\theta) + \int \nabla_\theta U(\theta, \theta') \, \phi(\theta' \mid \mu_t, \Sigma_t) \, \mathrm{d}\theta' \right] \phi(\theta \mid \mu_t, \Sigma) \, \mathrm{d}\theta \\
&= \frac{2}{n} \sum_{i=1}^n \big( y_i - \mathbb{E}_{\rho_t} f(x_i, \theta) \big) \, \mathbb{E}_{\rho_t} \nabla_\theta f(x_i, \theta)
\end{aligned}
$$

---

[4]In general, the Wasserstein gradient at a measure $\mu$ is an element of $L^2(\mu)$ and therefore is only defined almost everywhere; see Ambrosio et al. (2008, Chapter 8) for details.

and

$$
\begin{aligned}
\dot{\Sigma}_t &= -2\,\nabla_\Sigma \mathcal{L}(\rho_t)\,\Sigma_t - 2\,\Sigma_t\,\nabla_\Sigma \mathcal{L}(\rho_t) \\
&= -2\int \nabla_\theta V(\theta) \otimes (\theta - \mu_t)\,\phi(\theta \mid \mu_t, \Sigma_t)\,\mathrm{d}\theta \\
&\quad - 2\int\left[\int \nabla_\theta U(\theta,\theta') \otimes (\theta - \mu_t)\,\phi(\theta \mid \mu_t, \Sigma_t)\,\mathrm{d}\theta\right]\phi(\theta' \mid \mu_t, \Sigma_t)\,\mathrm{d}\theta' \\
&\quad - 2\int (\theta - \mu_t) \otimes \nabla_\theta V(\theta)\,\phi(\theta \mid \mu_t, \Sigma_t)\,\mathrm{d}\theta \\
&\quad - 2\int\left[\int (\theta - \mu_t) \otimes \nabla_\theta U(\theta,\theta')\,\phi(\theta \mid \mu_t, \Sigma_t)\,\mathrm{d}\theta\right]\phi(\theta' \mid \mu_t, \Sigma_t)\,\mathrm{d}\theta' \\
&= \frac{2}{n}\sum_{i=1}^{n}\big(y_i - \mathbb{E}_{\rho_t} f(x_i,\theta)\big)\big(\mathbb{E}_{\rho_t}[\nabla_\theta f(x_i,\theta) \otimes (\theta - \mu_t)] + \mathbb{E}_{\rho_t}[(\theta - \mu_t) \otimes \nabla_\theta f(x_i,\theta)]\big)\,.
\end{aligned}
$$

This completes the derivation. $\qquad\square$

Although Theorem 2 derives equations for the Bures–Wasserstein gradient flow, the expressions involve expectations which must also be computed. We now proceed to show that these expectations can be computed in closed form for ReLU activations, which allows for exact implementation in software. The following derivations are tedious, but the resulting equations are readily programmed.

We need to compute

$$
\mathbb{E}_\rho f(x_i,\theta) = \mathbb{E}_\rho[\omega\,⌿(\beta^\top x_i)]\,, \qquad \mathbb{E}_\rho \nabla_\theta f(x_i,\theta) = \mathbb{E}_\rho\begin{bmatrix} ⌿(\beta^\top x_i) \\ \omega\,⌿'(\beta^\top x_i)\,x_i \end{bmatrix}, \qquad \text{and}
$$

$$
\mathbb{E}_\rho[\nabla_\theta f(x_i,\theta) \otimes \theta] = \mathbb{E}_\rho\begin{bmatrix} \omega\,⌿(\beta^\top x_i) & ⌿(\beta^\top x_i)\,\beta^\top \\ \omega^2\,⌿'(\beta^\top x_i)\,x_i & \omega\,⌿'(\beta^\top x_i)\,x_i \otimes \beta \end{bmatrix}.
$$

Basically, we need to compute, for each $1 \le i \le n$,

$$
\begin{aligned}
A_i &= \mathbb{E}_\rho\,⌿(\beta^\top x_i)\,, & B_i &= \mathbb{E}_\rho[\omega\,⌿'(\beta^\top x_i)]\,, \\
C_i &= \mathbb{E}_\rho[\omega\,⌿(\beta^\top x_i)]\,, & D_i &= \mathbb{E}_\rho[\omega^2\,⌿'(\beta^\top x_i)]\,,
\end{aligned}
$$

and for each $1 \le j \le d$,

$$
P_{i,j} = \mathbb{E}_\rho[⌿(\beta^\top x_i)\,\beta_j]\,, \qquad Q_{i,j} = \mathbb{E}_\rho[\omega\,⌿'(\beta^\top x_i)\,\beta_j]\,.
$$

Then we can express

$$
\mathbb{E}_\rho \nabla_\theta f(x_i,\theta) = \begin{bmatrix} A_i \\ B_i x_i \end{bmatrix}, \qquad \mathbb{E}_\rho f(x_i,\theta) = C_i\,,
$$

and

$$
\mathbb{E}_\rho[\nabla_\theta f(x_i,\theta) \otimes (\theta - \mu)] = \begin{bmatrix} C_i & [P_{i,j}]_{1 \le j \le d} \\ D_i x_i & x_i \otimes [Q_{i,j}]_{1 \le j \le d} \end{bmatrix} - \begin{bmatrix} A_i \\ B_i x_i \end{bmatrix} \otimes \mu\,.
$$

Therefore, the update rule looks like

$$
\mu_{t+1} = \mu_t + \eta g_t \qquad \text{and} \qquad \Sigma_{t+1} = \Sigma_t + \eta G_t
$$

where

$$
g_t = \frac{2}{n}\sum_{i=1}^{n}(y_i - C_i)\begin{bmatrix} A_i \\ B_i x_i \end{bmatrix}
$$

and

$$G_t = \frac{2}{n} \sum_{i=1}^{n} (y_i - C_i) \left\{ \begin{bmatrix} C_i & [P_{i,j}]_{1 \le j \le d} \\ D_i x_i & x_i \otimes [Q_{i,j}]_{1 \le j \le d} \end{bmatrix} - \begin{bmatrix} A_i \\ B_i x_i \end{bmatrix} \otimes \mu \right\}$$
$$+ \frac{2}{n} \sum_{i=1}^{n} (y_i - C_i) \left\{ \begin{bmatrix} C_i & D_i x_i^\top \\ [P_{i,j}]_{1 \le j \le d} & [Q_{i,j}]_{1 \le j \le d} \otimes x_i \end{bmatrix} - \mu \otimes \begin{bmatrix} A_i \\ B_i x_i \end{bmatrix} \right\}.$$

In addition, the objective function

$$\mathcal{L}(\rho_t) = \frac{1}{n} \sum_{i=1}^{n} (y_i - C_i)^2.$$

Let us first compute a universal rule. Let $X = \omega$, $Y = \beta^\top x_i$ and $Z = \beta_j$. We have

$$\begin{bmatrix} X \\ Y \\ Z \end{bmatrix} \sim \mathcal{N} \left( \begin{bmatrix} \mu_1 \\ \mu_2 \\ \mu_3 \end{bmatrix}, \begin{bmatrix} \sigma_1^2 & \rho_{1,2}\sigma_1\sigma_2 & \rho_{1,3}\sigma_1\sigma_3 \\ \rho_{1,2}\sigma_1\sigma_2 & \sigma_2^2 & \rho_{2,3}\sigma_2\sigma_3 \\ \rho_{1,3}\sigma_1\sigma_3 & \rho_{2,3}\sigma_2\sigma_3 & \sigma_3^2 \end{bmatrix} \right), \tag{11}$$

where

$$\begin{bmatrix} \mu_1 \\ \mu_2 \\ \mu_3 \end{bmatrix} = \begin{bmatrix} \mu^\omega \\ x_i^\top \mu^\beta \\ e_j^\top \mu^\beta \end{bmatrix}, \qquad \begin{bmatrix} \sigma_1^2 & \rho_{1,2}\sigma_1\sigma_2 & \rho_{1,3}\sigma_1\sigma_3 \\ \rho_{1,2}\sigma_1\sigma_2 & \sigma_2^2 & \rho_{2,3}\sigma_2\sigma_3 \\ \rho_{1,3}\sigma_1\sigma_3 & \rho_{2,3}\sigma_2\sigma_3 & \sigma_3^2 \end{bmatrix} = \begin{bmatrix} (\sigma^\omega)^2 & \Sigma^{\omega,\beta} x_i & \Sigma^{\omega,\beta} e_j \\ x_i^\top \Sigma^{\beta,\omega} & x_i^\top \Sigma^\beta x_i & x_i^\top \Sigma^\beta e_j \\ e_j^\top \Sigma^{\beta,\omega} & e_j^\top \Sigma^\beta x_i & e_j^\top \Sigma^\beta e_j \end{bmatrix}.$$

We know that

$$\mathsf{cov}(X - \alpha Y, Y) = \rho_{1,2}\sigma_1\sigma_2 - \alpha\sigma_2^2 = 0 \qquad \text{when} \qquad \alpha = \rho_{1,2}\frac{\sigma_1}{\sigma_2}, \qquad \text{and}$$

$$\mathsf{cov}\big(Z - \beta(X - \alpha Y) - \gamma Y, Y\big) = \rho_{2,3}\sigma_2\sigma_3 - \gamma\sigma_2^2 = 0 \qquad \text{when} \qquad \gamma = \rho_{2,3}\frac{\sigma_3}{\sigma_2}, \qquad \text{and}$$

$$\mathsf{cov}\big(Z - \beta(X - \alpha Y) - \gamma Y, X - \alpha Y\big) = \mathsf{cov}(Z, X - \alpha Y) - \beta\,\mathsf{var}(X - \alpha Y) = 0$$

when

$$\beta = \frac{\rho_{1,3}\sigma_1\sigma_3 - \alpha\rho_{2,3}\sigma_2\sigma_3}{\sigma_1^2 - 2\alpha\rho_{1,2}\sigma_1\sigma_2 + \alpha^2\sigma_2^2}.$$

Therefore $X - \alpha Y$, $Y$, and $Z - \beta(X - \alpha Y) - \gamma Y$ are mutually independent. We first compute $A_i$ and $B_i$. By direct computation,

$$A_i = \mathbb{E}\max\{Y, 0\} = \mu_2 + \sigma_2\,\mathbb{E}\max\left\{\frac{Y - \mu_2}{\sigma_2}, -\frac{\mu_2}{\sigma_2}\right\}$$
$$= \mu_2 + \sigma_2 \int_{-\mu_2/\sigma_2}^{\infty} \frac{x}{\sqrt{2\pi}} e^{-x^2/2}\,\mathrm{d}x - \mu_2\,\Phi\big(-\frac{\mu_2}{\sigma_2}\big)$$
$$= \mu_2 - \sigma_2 \int_{-\mu_2/\sigma_2}^{\infty} \frac{1}{\sqrt{2\pi}}\,\mathrm{d}e^{-x^2/2} - \mu_2\,\Phi\big(-\frac{\mu_2}{\sigma_2}\big) = \mu_2 + \sigma_2\,\phi\big(-\frac{\mu_2}{\sigma_2}\big) - \mu_2\,\Phi\big(-\frac{\mu_2}{\sigma_2}\big)$$

and

$$B_i = \mathbb{E}[X\mathbb{1}_{Y>0}] = \mathbb{E}[(X - \alpha Y)\mathbb{1}_{Y>0}] + \mathbb{E}[\alpha Y\mathbb{1}_{Y>0}]$$
$$= \mathbb{E}[X - \alpha Y]\,\mathbb{P}(Y > 0) + \alpha\,\mathbb{E}\max\{Y, 0\} = E_i F_i + \alpha A_i,$$

where we further define

$$E_i := \mathbb{E}[X - \alpha Y] = \mu_1 - \alpha\mu_2 \qquad \text{and} \qquad F_i := \mathbb{P}(Y > 0) = 1 - \Phi\big(-\frac{\mu_2}{\sigma_2}\big).$$

We then compute

$$C_i = \mathbb{E}[X \max\{Y, 0\}] = \mathbb{E}[(X - \alpha Y) \max\{Y, 0\}] + \mathbb{E}[\alpha Y \max\{Y, 0\}]$$
$$= \mathbb{E}[X - \alpha Y] \, \mathbb{E}[\max\{Y, 0\}] + \alpha \, \mathbb{E}[Y \max\{Y, 0\}] = A_i E_i + \alpha G_i$$

where we define, for $Y \sim \mathcal{N}(\mu_2, \sigma_2^2)$,

$$G_i := \mathbb{E}[Y \max\{Y, 0\}] = \mathbb{E}[Y^2 \mathbb{1}_{Y>0}] = \int_0^\infty \frac{x^2}{\sqrt{2\pi}\sigma_2} \, e^{-(x-\mu_2)^2/(2\sigma_2^2)} \, \mathrm{d}x$$

$$= \int_{-\mu_2/\sigma_2}^\infty \frac{(\mu_2 + \sigma_2 y)^2}{\sqrt{2\pi}} \, e^{-y^2/2} \, \mathrm{d}y \qquad \text{by change of variable} \qquad y = \frac{x - \mu_2}{\sigma_2}$$

$$= \mu_2^2 \int_{-\mu_2/\sigma_2}^\infty \frac{1}{\sqrt{2\pi}} \, e^{-y^2/2} \, \mathrm{d}y + 2\mu_2\sigma_2 \int_{-\mu_2/\sigma_2}^\infty \frac{y}{\sqrt{2\pi}} \, e^{-y^2/2} \, \mathrm{d}y + \int_{-\mu_2/\sigma_2}^\infty \frac{\sigma_2^2 y^2}{\sqrt{2\pi}} \, e^{-y^2/2} \, \mathrm{d}y$$

$$= \mu_2^2 \left[1 - \Phi(-\mu_2/\sigma_2)\right] + 2\mu_2\sigma_2 \, \phi(-\mu_2/\sigma_2)$$

$$\qquad - \sigma_2^2 \frac{y}{\sqrt{2\pi}} \, e^{-y^2/2} \Big|_{-\mu_2/\sigma_2}^\infty + \sigma_2^2 \int_{-\mu_2/\sigma_2}^\infty \frac{1}{\sqrt{2\pi}} \, e^{-y^2/2} \, \mathrm{d}y$$

$$= (\mu_2^2 + \sigma_2^2) \left[1 - \Phi(-\mu_2/\sigma_2)\right] + \mu_2\sigma_2 \, \phi(-\mu_2/\sigma_2) \,.$$

We also need to compute

$$D_i = \mathbb{E}[X^2 \mathbb{1}_{Y>0}] = \mathbb{E}[(X - \alpha Y)^2 \mathbb{1}_{Y>0}] + \alpha^2 \, \mathbb{E}[Y^2 \mathbb{1}_{Y>0}] + 2\alpha \, \mathbb{E}[(X - \alpha Y) Y \mathbb{1}_{Y>0}]$$
$$= F_i H_i + \alpha^2 G_i + 2\alpha A_i E_i$$

where

$$H_i := \mathbb{E}[(X - \alpha Y)^2] = \sigma_1^2 - 2\alpha\rho_{1,2}\sigma_1\sigma_2 + \alpha^2\sigma_2^2 + (\mu_1 - \alpha\mu_2)^2 \,.$$

Then, we compute

$$P_{i,j} = \mathbb{E}[Z \max\{Y, 0\}] = \mathbb{E}[(Z - \gamma Y) \max\{Y, 0\}] + \mathbb{E}[\gamma Y \max\{Y, 0\}]$$
$$= \mathbb{E}[Z - \gamma Y] \, \mathbb{E}[\max\{Y, 0\}] + \gamma \, \mathbb{E}[Y \max\{Y, 0\}]$$
$$= A_i M_{i,j} + \gamma G_i \,,$$

where we let

$$M_{i,j} := \mathbb{E}[Z - \gamma Y] = \mu_3 - \gamma\mu_2 \,.$$

Finally, we compute

$$Q_{i,j} = \mathbb{E}[X \mathbb{1}_{Y>0} Z] = \mathbb{E}[(X - \alpha Y + \alpha Y) \mathbb{1}_{Y>0} (Z - \beta (X - \alpha Y) - \gamma Y + \beta (X - \alpha Y) + \gamma Y)]$$
$$= \mathbb{E}[(X - \alpha Y) \mathbb{1}_{Y>0} (Z - \beta (X - \alpha Y) - \gamma Y)] + \mathbb{E}[\alpha Y \mathbb{1}_{Y>0} (Z - \beta (X - \alpha Y) - \gamma Y)]$$
$$\qquad + \mathbb{E}[(X - \alpha Y) \mathbb{1}_{Y>0} \beta (X - \alpha Y)] + \mathbb{E}[\alpha Y \mathbb{1}_{Y>0} \beta (X - \alpha Y)]$$
$$\qquad + \mathbb{E}[(X - \alpha Y) \mathbb{1}_{Y>0} \gamma Y] + \mathbb{E}[\alpha Y \mathbb{1}_{Y>0} \gamma Y]$$
$$= E_i F_i N_{i,j} + \alpha A_i N_{i,j} + \beta F_i H_i + \alpha\beta A_i E_i + \gamma A_i E_i + \alpha\gamma G_i \,,$$

where we define

$$N_{i,j} := \mathbb{E}[Z - \beta (X - \alpha Y) - \gamma Y] = \mu_3 - \beta\mu_1 + \alpha\beta\mu_2 - \gamma\mu_2 \,.$$

We have provided explicit expressions for all of the terms.

### A.1.2 Gaussian mixtures

Consider a $K$-component Gaussian mixture distribution $\rho_\nu$ parameterized by $\nu$:

$$\rho_\nu = \frac{1}{K} \sum_{k=1}^K \mathcal{N}(\mu^{(k)}, \Sigma^{(k)}) \,, \qquad \text{where} \qquad \nu = \frac{1}{K} \sum_{k=1}^K \delta_{(\mu^{(k)}, \Sigma^{(k)})} \,.$$

Here $\nu$ is a discrete probability measure over $\mathbb{R}^d \times \mathbf{S}_{++}^d$. We start by deriving the Gaussian mixture gradient flow for a general loss $\mathcal{L}$.

**Theorem 3.** *Let $\mathcal{L}$ be a functional over the Wasserstein space. The Gaussian mixture gradient flow $(\nu_t)_{t\geq 0}$ for minimizing $\mathcal{L}$ initialized at a distribution $\nu_0 = K^{-1}\sum_{k=1}^{K}\delta_{(\mu_0^{(k)}, \Sigma_0^{(k)})}$ with $K$ atoms is given by $\nu_t = K^{-1}\sum_{k=1}^{K}\delta_{(\mu_t^{(k)}, \Sigma_t^{(k)})}$, $t \geq 0$, where for each $k \in [K]$, the dynamics of $(\mu_t^{(k)})_{t\geq 0}$ and $(\Sigma_t^{(k)})_{t\geq 0}$ are governed by the ODE system*

$$\begin{aligned}
\dot{\mu}_t^{(k)} &= -\mathbb{E}_{\mathcal{N}(\mu_t^{(k)}, \Sigma_t^{(k)})} \nabla \delta\mathcal{L}(\rho_{\nu_t}), \\
\dot{\Sigma}_t^{(k)} &= -\Sigma_t^{(k)} \mathbb{E}_{\mathcal{N}(\mu_t^{(k)}, \Sigma_t^{(k)})} \nabla^2 \delta\mathcal{L}(\rho_{\nu_t}) - \mathbb{E}_{\mathcal{N}(\mu_t^{(k)}, \Sigma_t^{(k)})} \nabla^2 \delta\mathcal{L}(\rho_{\nu_t}) \Sigma_t^{(k)}.
\end{aligned}$$

*Here, $\delta\mathcal{L} = \delta_\rho\mathcal{L}$ refers to the first variation of $\mathcal{L}$ (see Santambrogio, 2015, Chapter 7).*

*Proof.* We refer to Lambert et al. (2022, Appendix F) for background. We calculate the first variation of $\nu \mapsto \mathcal{L}(\rho_\nu)$ in terms of the first variation of $\mathcal{L}$: note that for any $\delta > 0$ and any measure $\mathcal{X}$ satisfying $\int_{\mathsf{BW}(\mathbb{R}^d)} \mathrm{d}\mathcal{X} = 0$ and such that $\nu + \delta\mathcal{X}$ for sufficiently small $\delta > 0$, if $\xi = (\mu, \Sigma)$ and $p_\xi$ denotes $\mathcal{N}(\mu, \Sigma)$, we have

$$\mathcal{L}(\rho_{\nu+\delta\mathcal{X}}) - \mathcal{L}(\rho_\nu) = \int \delta\mathcal{L}(\rho_\nu)\,\mathrm{d}(\rho_{\nu+\delta\mathcal{X}} - \rho_\nu) + o(\delta) = \int \left[\int \delta\mathcal{L}(\rho_\nu)\,\mathrm{d}p_\xi\right] \delta\mathcal{X}(\mathrm{d}\xi) + o(\delta),$$

which, by the definition of the first variation, shows that

$$\delta_\nu\mathcal{L}(\rho_\nu) : \xi \mapsto \int \delta\mathcal{L}(\rho_\nu)\,\mathrm{d}p_\xi.$$

Since the Gaussian mixture gradient flow is, by definition, a Wasserstein gradient flow over the Bures–Wasserstein space, the particle interpretation of Wasserstein gradient flows shows that each $(\mu_t^{(k)}, \Sigma_t^{(k)})$ evolves by the Bures–Wasserstein gradient of $\delta_\nu\mathcal{L}(\rho_{\nu_t})$. It follows from Lambert et al. (2022, Appendix C) that the Gaussian mixture flow takes the claimed form.

Alternatively, we can derive the gradient flow more explicitly. Noting that $\xi = (\mu, \Sigma)$, we have

$$\begin{aligned}
\nabla_\mu \delta_\nu\mathcal{L}(\rho_\nu)(\xi) &= \int \delta\mathcal{L}(\rho_\nu)\,\nabla_\mu\phi(\theta \mid \mu, \Sigma)\,\mathrm{d}\theta = -\int \delta\mathcal{L}(\rho_\nu)\,\nabla_\theta\phi(\theta \mid \mu, \Sigma)\,\mathrm{d}\theta \\
&= \int \nabla_\theta\delta\mathcal{L}(\rho_\nu)\,\phi(\theta \mid \mu, \Sigma)\,\mathrm{d}\theta
\end{aligned}$$

and

$$\begin{aligned}
\nabla_\Sigma \delta_\nu\mathcal{L}(\rho_\nu)(\xi) &= \int \delta\mathcal{L}(\rho_\nu)\,\nabla_\Sigma\phi(\theta \mid \mu, \Sigma)\,\mathrm{d}\theta = \frac{1}{2}\int \delta\mathcal{L}(\rho_\nu)\,\nabla_\theta^2\phi(\theta \mid \mu, \Sigma)\,\mathrm{d}\theta \\
&= \frac{1}{2}\int \nabla_\theta^2\delta\mathcal{L}(\rho_\nu)\,\phi(\theta \mid \mu, \Sigma)\,\mathrm{d}\theta,
\end{aligned}$$

where we used the expressions in Appendix A.3. Recalling (3), it completes the derivation. $\qquad\square$

With Theorem 3 in hand, we can now prove Theorem 1.

*Proof of Theorem 1.* Consider the loss function

$$\ell(\boldsymbol{\mu}, \boldsymbol{C}) \equiv \ell(\mu_1, \ldots, \mu_K, C_1, \ldots, C_K) := \mathcal{L}(\rho_\nu)$$

where

$$\nu = \frac{1}{K}\sum_{k=1}^{K} \delta_{(\mu_k, \Sigma_k)} \quad \text{with} \quad \Sigma_k = C_k C_k^\top.$$

Note that for any $\Delta \in \mathbb{R}^d$ and any $\varepsilon > 0$, we have

$$\lim_{\varepsilon \to 0} \frac{\ell(\mu_1 + \varepsilon\Delta, \mu_2, \ldots, \mu_K, C_1, \ldots, C_K) - \ell(\mu_1, \ldots, \mu_K, C_1, \ldots, C_K)}{\varepsilon} = \langle \nabla_{\mu_1}\ell(\boldsymbol{\mu}, \boldsymbol{C}), \Delta \rangle.$$

The left-hand side of the above equation can also be expressed as

$$\lim_{\varepsilon \to 0} \frac{\mathcal{L}(\rho_{\nu_\varepsilon}) - \mathcal{L}(\rho_\nu)}{\varepsilon} \qquad \text{where} \qquad \nu_\varepsilon := \frac{1}{K} \delta_{(\mu_1 + \varepsilon\Delta, \Sigma_1)} + \frac{1}{K} \sum_{k=2}^{K} \delta_{(\mu_k, \Sigma_k)}.$$

By the definition of first variation, we know that

$$\mathcal{L}(\rho_{\nu_\varepsilon}) - \mathcal{L}(\rho_\nu) = \int \delta\mathcal{L}(\rho_\nu) \, \mathrm{d}(\rho_{\nu_\varepsilon} - \rho_\nu) + o(\varepsilon)$$

$$= \int \left[ \int \delta\mathcal{L}(\rho_\nu) \, \mathrm{d}p_\xi \right] \varepsilon\mathcal{X}(\mathrm{d}\xi) + o(\varepsilon) \qquad \text{where} \qquad \mathcal{X} = \nu_\varepsilon - \nu$$

$$= \frac{1}{K} \left[ \int \delta\mathcal{L}(\rho_\nu) \, \mathrm{d}p_{(\mu_1 + \varepsilon\Delta, \Sigma_1)} - \int \delta\mathcal{L}(\rho_\nu) \, \mathrm{d}p_{(\mu_1, \Sigma_1)} \right] + o(\varepsilon).$$

Taking the above three relations collectively yields

$$\langle \nabla_{\mu_1} \ell(\boldsymbol{\mu}, \boldsymbol{C}), \Delta \rangle = \frac{1}{K} \left\langle \nabla_{\mu_1} \int \delta\mathcal{L}(\rho_\nu) \, \mathrm{d}p_{(\mu_1, \Sigma_1)}, \Delta \right\rangle.$$

Since the above equation holds for any $\Delta \in \mathbb{R}^d$, we know that

$$\nabla_{\mu_1} \ell(\boldsymbol{\mu}, \boldsymbol{C}) = \frac{1}{K} \nabla_{\mu_1} \int \delta\mathcal{L}(\rho_\nu) \, \mathrm{d}p_{(\mu_1, \Sigma_1)} = \frac{1}{K} \mathbb{E}_{\mathcal{N}(\mu_1, \Sigma_1)} \nabla\delta\mathcal{L}(\rho_{\nu_t}),$$

where we used the expressions in Appendix A.3 and integration by parts. Therefore the Euclidean gradient flow w.r.t. $\mu^{(k)}$ is given by

$$\dot{\mu}_t^{(k)} = -\frac{1}{K} \mathbb{E}_{\mathcal{N}(\mu_t^{(k)}, \Sigma_t^{(k)})} \nabla\delta\mathcal{L}(\rho_{\nu_t}). \tag{12}$$

Similarly, we can show that

$$\nabla_{\Sigma_1} \ell(\boldsymbol{\mu}, \boldsymbol{C}) = \frac{1}{K} \nabla_{\Sigma_1} \int \delta\mathcal{L}(\rho_\nu) \, \mathrm{d}p_{(\mu_1, \Sigma_1)} = \frac{1}{2K} \mathbb{E}_{\mathcal{N}(\mu_1, \Sigma_1)} \nabla^2\delta\mathcal{L}(\rho_{\nu_t}),$$

which then leads to

$$\nabla_{C_1} \ell(\boldsymbol{\mu}, \boldsymbol{C}) = 2 \nabla_{\Sigma_1} \ell(\boldsymbol{\mu}, \boldsymbol{C}) \, C_1 = \frac{1}{K} \mathbb{E}_{\mathcal{N}(\mu_1, \Sigma_1)} \nabla^2\delta\mathcal{L}(\rho_{\nu_t}) \, C_1.$$

Hence the Euclidean gradient flow w.r.t. $C^{(k)}$ is given by

$$\dot{C}_t^{(k)} = -\frac{1}{K} \mathbb{E}_{\mathcal{N}(\mu_t^{(k)}, \Sigma_t^{(k)})} \nabla^2\delta\mathcal{L}(\rho_{\nu_t}) \, C_t^{(k)},$$

therefore $\Sigma_t^{(k)} = C_t^{(k)} C_t^{(k)\top}$ satisfies

$$\dot{\Sigma}_t^{(k)} = C_t^{(k)} \dot{C}_t^{(k)\top} + \dot{C}_t^{(k)} C_t^{(k)\top}$$

$$= -\frac{1}{K} \left[ \mathbb{E}_{\mathcal{N}(\mu_t^{(k)}, \Sigma_t^{(k)})} \nabla^2\delta\mathcal{L}(\rho_{\nu_t}) \right] \Sigma_t^{(k)} - \frac{1}{K} \Sigma_t^{(k)} \left[ \mathbb{E}_{\mathcal{N}(\mu_t^{(k)}, \Sigma_t^{(k)})} \nabla^2\delta\mathcal{L}(\rho_{\nu_t}) \right]. \tag{13}$$

By comparing (12) and (13) with the ODE system in Theorem 3, we can see that they are equivalent up to a scaling factor of $K$. $\qquad\square$

Then, we specialize to the objective function

$$\min_\nu \mathcal{L}(\rho_\nu) := \frac{1}{n} \sum_{i=1}^{n} \left( y_i - \int f(x_i, \theta) \, \rho_\nu(\mathrm{d}\theta) \right)^2.$$

**Theorem 4.** *The Gaussian mixture gradient flow $(\nu_t)_{t \geq 0}$ for minimizing $\mathcal{L}$ initialized at a distribution $\nu_0 = K^{-1} \sum_{k=1}^K \delta_{(\mu_0^{(k)}, \Sigma_0^{(k)})}$ with $K$ atoms is given by $\nu_t = K^{-1} \sum_{k=1}^K \delta_{(\mu_t^{(k)}, \Sigma_t^{(k)})}$, $t \geq 0$, where for each $k \in [K]$, the dynamics of $(\mu_t^{(k)})_{t \geq 0}$ and $(\Sigma_t^{(k)})_{t \geq 0}$ are governed by the ODE system*

$$
\begin{aligned}
\dot{\mu}_t^{(k)} &= \frac{2}{n} \sum_{i=1}^n \big(y_i - \mathbb{E}_{\rho_{\nu_t}} f(x_i, \theta)\big) \, \mathbb{E}_{\mathcal{N}(\mu_t^{(k)}, \Sigma_t^{(k)})} \nabla_\theta f(x_i, \theta) \,, \\
\dot{\Sigma}_t^{(k)} &= \frac{2}{n} \sum_{i=1}^n \big(y_i - \mathbb{E}_{\rho_{\nu_t}} f(x_i, \theta)\big) \\
&\qquad \times \mathbb{E}_{\mathcal{N}(\mu_t^{(k)}, \Sigma_t^{(k)})} [\nabla_\theta f(x_i, \theta) \otimes (\theta - \mu_t^{(k)}) + (\theta - \mu_t^{(k)}) \otimes \nabla_\theta f(x_i, \theta)] \,.
\end{aligned}
$$

*Proof.* The first variation is given by

$$
\delta \mathcal{L}(\rho) : \theta \mapsto V(\theta) + \int U(\theta, \theta') \, \rho(\mathrm{d}\theta') \,,
$$

where $U$ and $V$ are as in (7). Then, we can apply Theorem 3, which is seen to yield the desired equations after substituting in the definitions of $U$ and $V$ and performing integration by parts. $\qquad \square$

As before, we write out more explicit expressions for the gradient flow. Thankfully, we can reuse our previous calculations. Compared to Appendix A.1.1, we only need to compute

$$
C_i' := \mathbb{E}_{\rho_\nu} f(x_i, \theta) = \mathbb{E}_{\rho_\nu} [\omega \, \diagup(\beta^\top x_i)] = \frac{1}{K} \sum_{k=1}^K \mathbb{E}_{\mathcal{N}(\mu^{(k)}, \Sigma^{(k)})} [\omega \, \diagup(\beta^\top x_i)] = \frac{1}{K} \sum_{k=1}^K C_i^{(k)} \,.
$$

Then, the update becomes

$$
\mu_{t+1}^{(k)} = \mu_t^{(k)} + \eta g_t^{(k)} \qquad \text{and} \qquad \Sigma_{t+1}^{(k)} = \Sigma_t^{(k)} + \eta G_t^{(k)}
$$

where

$$
g_t^{(k)} = \frac{2}{n} \sum_{i=1}^n (y_i - C_i') \begin{bmatrix} A_i \\ B_i x_i \end{bmatrix}
$$

and

$$
\begin{aligned}
G_t^{(k)} &= \frac{2}{n} \sum_{i=1}^n (y_i - C_i') \left\{ \begin{bmatrix} C_i & [P_{i,j}]_{1 \leq j \leq d} \\ D_i x_i & x_i \otimes [Q_{i,j}]_{1 \leq j \leq d} \end{bmatrix} - \begin{bmatrix} A_i \\ B_i x_i \end{bmatrix} \otimes \mu \right\} \\
&\quad + \frac{2}{n} \sum_{i=1}^n (y_i - C_i') \left\{ \begin{bmatrix} C_i & D_i x_i^\top \\ [P_{i,j}]_{1 \leq j \leq d} & [Q_{i,j}]_{1 \leq j \leq d} \otimes x_i \end{bmatrix} - \mu \otimes \begin{bmatrix} A_i \\ B_i x_i \end{bmatrix} \right\} .
\end{aligned}
$$

## A.2 Multi-class classification

While our implementation relies on PyTorch's Automatic Differentiation engine, we present here exact computations of gradients for multi-class classification. Their complexity indicates that it is largely preferable to employ automatic differentiation and that a study of first-order optimality conditions appears challenging.

For this setting, we focus on the parametrization as described in Subsection 4.2, that is, $\beta \sim \mathcal{N}(\mu^\beta, \Sigma)$ and $\mathbb{E}[\omega \mid \beta] = U\beta + v$, where we apply the Euclidean gradient flow to the parameters $\mu^\beta$, $U$, $v$, and the square root of $\Sigma$. To simplify the notation, we write $\mu = \mu^\beta$. (The sparse parametrization in Subsection 4.2 corresponds to further restricting $\Sigma$ to be diagonal.)

We first compute the gradients w.r.t. $U$ and $v$. Write

$$
U = \begin{bmatrix} u_1^\top \\ \vdots \\ u_L^\top \end{bmatrix} \qquad \text{and} \qquad v = \begin{bmatrix} v_1 \\ \vdots \\ v_L \end{bmatrix} .
$$

The following expectations are understood to be taken over the Gaussian mixture. We have

$$\mathbb{E}\nabla_\beta\big(\phi(\beta^\top x)\,u^\top\beta\big) = \mathbb{E}[\nabla_\beta\,\phi(\beta^\top x)\,u^\top\beta + \phi(\beta^\top x)\,u]$$
$$= \mathbb{E}[u^\top\beta\,\phi'(\beta^\top x)\,\beta] + \mathbb{E}[\phi(\beta^\top x)]\,u\,.$$

The loss function is

$$\mathcal{L}(\rho) = -\frac{1}{n}\sum_{i=1}^{n}\Big\{\sum_{\ell=1}^{L}\mathbb{1}_{y_i=\ell}\int f_\ell(x_i,\theta)\,\rho(\mathrm{d}\theta) - \log\Big[1 + \sum_{\ell=1}^{L}\exp\int f_\ell(x_i,\theta)\,\rho(\mathrm{d}\theta)\Big]\Big\}$$

where

$$\int f_\ell(x_i,\theta)\,\rho(\mathrm{d}\theta) = \mathbb{E}[\phi(\beta^\top x)\,u_\ell^\top\beta] + v_\ell\,\mathbb{E}\,\phi(\beta^\top x)\,.$$

We can compute

$$\nabla_{u_\ell}\mathcal{L}(\rho) = -\frac{1}{n}\sum_{i=1}^{n}\Big\{\mathbb{1}_{y_i=\ell} - \frac{\exp\int f_\ell(x_i,\theta)\,\rho(\mathrm{d}\theta)}{1 + \sum_{\ell'=1}^{L}\exp\int f_{\ell'}(x_i,\theta)\,\rho(\mathrm{d}\theta)}\Big\}\,\mathbb{E}[\phi(\beta^\top x)\,\beta]\,,$$

$$\nabla_{v_\ell}\mathcal{L}(\rho) = -\frac{1}{n}\sum_{i=1}^{n}\Big\{\mathbb{1}_{y_i=\ell} - \frac{\exp\int f_\ell(x_i,\theta)\,\rho(\mathrm{d}\theta)}{1 + \sum_{\ell'=1}^{L}\exp\int f_{\ell'}(x_i,\theta)\,\rho(\mathrm{d}\theta)}\Big\}\,\mathbb{E}\,\phi(\beta^\top x)\,.$$

Next, we can compute

$$\nabla_\mu\mathcal{L}(\rho) = -\frac{1}{n}\sum_{i=1}^{n}\Big\{\sum_{\ell=1}^{L}\mathbb{1}_{y_i=\ell}\nabla_\mu\int f_\ell(x_i,\theta)\,\rho(\mathrm{d}\theta)$$
$$-\frac{\sum_{\ell=1}^{L}\exp(\int f_\ell(x_i,\theta)\,\rho(\mathrm{d}\theta))\,\nabla_\mu\int f_\ell(x_i,\theta)\,\rho(\mathrm{d}\theta)}{1 + \sum_{\ell=1}^{L}\exp\int f_\ell(x_i,\theta)\,\rho(\mathrm{d}\theta)}\Big\}$$
$$= -\frac{1}{n}\sum_{i=1}^{n}\sum_{\ell=1}^{L}\Big\{\mathbb{1}_{y_i=\ell} - \frac{\exp\int f_\ell(x_i,\theta)\,\rho(\mathrm{d}\theta)}{1 + \sum_{\ell=1}^{L}\exp\int f_\ell(x_i,\theta)\,\rho(\mathrm{d}\theta)}\Big\}\nabla_\mu\int f_\ell(x_i,\theta)\,\rho(\mathrm{d}\theta)$$

and similarly,

$$\nabla_\Sigma\mathcal{L}(\rho) = -\frac{1}{n}\sum_{i=1}^{n}\sum_{\ell=1}^{L}\Big\{\mathbb{1}_{y_i=\ell} - \frac{\exp\int f_\ell(x_i,\theta)\,\rho(\mathrm{d}\theta)}{1 + \sum_{\ell=1}^{L}\exp\int f_\ell(x_i,\theta)\,\rho(\mathrm{d}\theta)}\Big\}\nabla_\Sigma\int f_\ell(x_i,\theta)\,\rho(\mathrm{d}\theta)\,.$$

We have

$$\nabla_\mu\int f_\ell(x_i,\theta)\,\rho(\mathrm{d}\theta) = \nabla_\mu\mathbb{E}[\phi(\beta^\top x)\,u_\ell^\top\beta] + \nabla_\mu v_\ell\,\mathbb{E}\,\phi(\beta^\top x)$$
$$= \mathbb{E}\nabla_\beta\big(\phi(\beta^\top x)\,u_\ell^\top\beta\big) + v_\ell\,\mathbb{E}\nabla_\beta\,\phi(\beta^\top x)$$
$$= \mathbb{E}[\phi'(\beta^\top x)\,u_\ell^\top\beta]\,x + \mathbb{E}[\phi(\beta^\top x)]\,u_\ell + v_\ell\,\mathbb{E}[\phi'(\beta^\top x)]\,x\,.$$

We also have

$$\nabla_\Sigma\int f_\ell(x_i,\theta)\,\rho(\mathrm{d}\theta) = \nabla_\Sigma\mathbb{E}[\phi(\beta^\top x)\,u_\ell^\top\beta + v_\ell\,\phi(\beta^\top x)]\,,$$

where

$$\nabla_\Sigma\mathbb{E}[\phi(\beta^\top x)\,u_\ell^\top\beta] = \frac{1}{2}\,\mathbb{E}\big[\nabla_\beta[\phi(\beta^\top x)\,u_\ell^\top\beta]\otimes(\beta-\mu)\big]\,\Sigma^{-1}$$
$$= \frac{1}{2}\,\Sigma^{-1}\,\mathbb{E}\big[(\beta-\mu)\otimes\nabla_\beta[\phi(\beta^\top x)\,u_\ell^\top\beta]\big]$$
$$= \frac{1}{2}\,\mathbb{E}\big[\big(\phi'(\beta^\top x)\,u_\ell^\top\beta\,x + \phi(\beta^\top x)\,u_\ell\big)\otimes(\beta-\mu)\big]\,\Sigma^{-1}$$
$$= \frac{1}{2}\,\Sigma^{-1}\,\mathbb{E}\big[(\beta-\mu)\otimes\big(\phi'(\beta^\top x)\,u_\ell^\top\beta\,x + \phi(\beta^\top x)\,u_\ell\big)\big]$$

and

$$\nabla_\Sigma \mathbb{E}\, g(\beta^\top x) = \frac{1}{2}\, \mathbb{E}[\nabla_\beta\, g(\beta^\top x)\otimes(\beta-\mu)]\,\Sigma^{-1} = \frac{1}{2}\,\Sigma^{-1}\,\mathbb{E}[(\beta-\mu)\otimes\nabla_\beta\, g(\beta^\top x)]$$
$$= \frac{1}{2}\, \mathbb{E}[g'(\beta^\top x)\,x\otimes(\beta-\mu)]\,\Sigma^{-1} = \frac{1}{2}\,\Sigma^{-1}\,\mathbb{E}[(\beta-\mu)\otimes g'(\beta^\top x)\,x]\,.$$

Hence, we have

$$\dot\mu = -\nabla_\mu\mathcal{L}(\rho) = \frac{1}{n}\sum_{i=1}^{n}\sum_{\ell=1}^{L}\Big[\mathbb{1}_{y_i=\ell}-\frac{\exp\int f_\ell(x_i,\theta)\,\rho(\mathrm{d}\theta)}{1+\sum_{\ell=1}^{L}\exp\int f_\ell(x_i,\theta)\,\rho(\mathrm{d}\theta)}\Big]\nabla_\mu\int f_\ell(x_i,\theta)\,\rho(\mathrm{d}\theta)$$
$$= \frac{1}{n}\sum_{i=1}^{n}\sum_{\ell=1}^{L}\Big[\mathbb{1}_{y_i=\ell}-\frac{\exp\int f_\ell(x_i,\theta)\,\rho(\mathrm{d}\theta)}{1+\sum_{\ell=1}^{L-1}\exp\int f_\ell(x_i,\theta)\,\rho(\mathrm{d}\theta)}\Big]$$
$$\times\big[\mathbb{E}[g'(\beta^\top x)\,u_\ell^\top\beta]\,x + \mathbb{E}[g(\beta^\top x)]\,u_\ell + v_\ell\,\mathbb{E}[g'(\beta^\top x)]\,x\big]$$
$$= \frac{1}{n}\sum_{i=1}^{n}\sum_{\ell=1}^{L-1}\Big[\mathbb{1}_{y_i=\ell}-\frac{\exp\int f_\ell(x_i,\theta)\,\rho(\mathrm{d}\theta)}{1+\sum_{\ell=1}^{L-1}\exp\int f_\ell(x_i,\theta)\,\rho(\mathrm{d}\theta)}\Big]$$
$$\times\big[u_\ell^\top\,\mathbb{E}[g'(\beta^\top x_i)\,\beta]\,x_i + \mathbb{E}[g(\beta^\top x_i)]\,u_\ell + v_\ell\,\mathbb{E}[g'(\beta^\top x)]\,x\big]$$

and

$$\dot\Sigma = -2\,\nabla_\Sigma\mathcal{L}(\rho)\,\Sigma - 2\,\Sigma\,\nabla_\Sigma\mathcal{L}(\rho)$$
$$= \frac{1}{n}\sum_{i=1}^{n}\sum_{\ell=1}^{L-1}\Big[\mathbb{1}_{y_i=\ell}-\frac{\exp\int f_\ell(x_i,\theta)\,\rho(\mathrm{d}\theta)}{1+\sum_{\ell=1}^{L-1}\exp\int f_\ell(x_i,\theta)\,\rho(\mathrm{d}\theta)}\Big]$$
$$\times\Big\{\mathbb{E}\big[\big(g'(\beta^\top x_i)\,u_\ell^\top\beta\,x_i + g(\beta^\top x_i)\,u_\ell + g'(\beta^\top x_i)\,x_i\big)\otimes(\beta-\mu)\big]$$
$$+ \mathbb{E}\big[\big(g'(\beta^\top x_i)\,u_\ell^\top\beta\,x_i + g(\beta^\top x_i)\,u_\ell + g'(\beta^\top x_i)\,x_i\big)\otimes(\beta-\mu)\big]^\top\Big\}\,.$$

Let

$$A_i = \mathbb{E}\, g(\beta^\top x_i)\,,\qquad F_i = \mathbb{E}\, g'(\beta^\top x_i)\,,\qquad R_{i,j} = \mathbb{E}[g'(\beta^\top x_i)\,\beta_j]\,,$$

and

$$P_{i,j} = \mathbb{E}[g(\beta^\top x_i)\,\beta_j]\,,\qquad Q_{i,j,\ell} = \mathbb{E}[g'(\beta^\top x_i)\,u_\ell^\top\beta\,\beta_j]\,,\qquad S_{i,\ell} = \mathbb{E}[\omega_\ell\, g(\beta^\top x_i)]\,.$$

We have

$$\dot\mu = \frac{1}{n}\sum_{i=1}^{n}\sum_{\ell=1}^{L-1}\Big[\mathbb{1}_{y_i=\ell}-\frac{\exp S_{i,\ell}}{1+\sum_{\ell'=1}^{L-1}\exp S_{i,\ell'}}\Big]\big[u_\ell^\top P_i\,x_i + A_i\,u_\ell + v_\ell\,F_i\,x_i\big]$$

and

$$\dot\Sigma = \frac{1}{n}\sum_{i=1}^{n}\sum_{\ell=1}^{L-1}\Big[\mathbb{1}_{y_i=\ell}-\frac{\exp S_{i,\ell}}{1+\sum_{\ell'=1}^{L-1}\exp S_{i,\ell'}}\Big]$$
$$\times\big[(x_i\otimes Q_{i,\ell} + u_\ell\otimes P_i + x_i\otimes R_i - u_\ell^\top R_i\,x_i\otimes\mu - A_i\,u_\ell\otimes\mu - F_i\,x_i\otimes\mu)+$$
$$+ (x_i\otimes Q_{i,\ell} + u_\ell\otimes P_i + x_i\otimes R_i - u_\ell^\top R_i\,x_i\otimes\mu - A_i\,u_\ell\otimes\mu - F_i\,x_i\otimes\mu)^\top\big]\,.$$

We also have

$$\dot U = -\begin{bmatrix}\nabla_{u_1}\mathcal{L}(\rho)^\top\\ \vdots\\ \nabla_{u_{L-1}}\mathcal{L}(\rho)^\top\end{bmatrix} = \frac{1}{n}\sum_{i=1}^{n}\begin{bmatrix}\mathbb{1}_{y_i=1}-\frac{\exp S_{i,1}}{1+\sum_{\ell'=1}^{L-1}\exp S_{i,\ell'}}\\ \vdots\\ \mathbb{1}_{y_i=L-1}-\frac{\exp S_{i,L-1}}{1+\sum_{\ell'=1}^{L-1}\exp S_{i,\ell'}}\end{bmatrix}\otimes P_i$$

and

$$\dot{v} = -\begin{bmatrix} \nabla_{v_1}\mathcal{L}(\rho)^\top \\ \vdots \\ \nabla_{v_{L-1}}\mathcal{L}(\rho)^\top \end{bmatrix} = \frac{1}{n}\sum_{i=1}^n \begin{bmatrix} \mathbb{1}_{y_i=1} - \frac{\exp S_{i,1}}{1+\sum_{\ell'=1}^{L-1}\exp S_{i,\ell'}} \\ \vdots \\ \mathbb{1}_{y_i=L-1} - \frac{\exp S_{i,L-1}}{1+\sum_{\ell'=1}^{L-1}\exp S_{i,\ell'}} \end{bmatrix} A_i\,.$$

Next, we compute each quantity. As before, we consider (11), except now we define $X = u_\ell^\top \beta$. This time, we have

$$\begin{bmatrix} \mu_1 \\ \mu_2 \\ \mu_3 \end{bmatrix} = \begin{bmatrix} u_\ell^\top \mu \\ x_i^\top \mu \\ e_j^\top \mu \end{bmatrix}, \qquad \begin{bmatrix} \sigma_1^2 & \rho_{1,2}\sigma_1\sigma_2 & \rho_{1,3}\sigma_1\sigma_3 \\ \rho_{1,2}\sigma_1\sigma_2 & \sigma_2^2 & \rho_{2,3}\sigma_2\sigma_3 \\ \rho_{1,3}\sigma_1\sigma_3 & \rho_{2,3}\sigma_2\sigma_3 & \sigma_3^2 \end{bmatrix} = \begin{bmatrix} u_\ell^\top \\ x_i^\top \\ e_j^\top \end{bmatrix} \Sigma \begin{bmatrix} u_\ell & x_i & e_j \end{bmatrix}.$$

Then, as before, we have

$$A_i = \mathbb{E}[\max\{Y,0\}]\,, \qquad F_i = \mathbb{P}(Y > 0)\,, \qquad R_{i,j} = \mathbb{E}[Z\mathbb{1}_{Y>0}]\,,$$

and

$$P_{i,j} = \mathbb{E}[\max\{Y,0\}Z] \qquad \text{and} \qquad Q_{i,j,\ell} = \mathbb{E}[X\mathbb{1}_{Y>0}Z]\,.$$

The only new quantity to compute is

$$\begin{aligned} R_{i,j} = \mathbb{E}[Z\mathbb{1}_{Y>0}] &= \mathbb{E}[(Z-\gamma Y)\mathbb{1}_{Y>0}] + \mathbb{E}[\gamma Y \mathbb{1}_{Y>0}] \\ &= \mathbb{E}[Z-\gamma Y]\,\mathbb{P}(Y>0) + \gamma\,\mathbb{E}[\max\{Y,0\}] \\ &= F_i M_{i,j} + \gamma A_i \end{aligned}$$

and

$$S_{i,\ell} = \mathbb{E}[\omega_\ell \,\diagdown\!\!(\beta^\top x_i)] = \mathbb{E}[\diagdown\!\!(\beta^\top x_i)\,u_\ell^\top \beta] + v_\ell\,\mathbb{E}\,\diagdown\!\!(\beta^\top x_i) = C_i + v_\ell A_i\,.$$

## A.3 Auxiliary Gaussian calculus

Recall that $\phi(\cdot \mid \mu, \Sigma)$ is the density function of $\mathcal{N}(\mu, \Sigma)$. Then we can compute the gradient and Hessian w.r.t. the variable $\theta$:

$$\begin{aligned} \nabla_\theta \phi(\theta \mid \mu, \Sigma) &= \phi(\theta \mid \mu, \Sigma)\,\nabla_\theta\big(-\tfrac{1}{2}\theta^\top \Sigma^{-1}\theta + \theta^\top \Sigma^{-1}\mu - \tfrac{1}{2}\mu^\top \Sigma^{-1}\mu\big) \\ &= -\phi(\theta \mid \mu, \Sigma)\,\Sigma^{-1}(\theta - \mu)\,, \\ \nabla_\theta^2 \phi(\theta \mid \mu, \Sigma) &= -\Sigma^{-1}(\theta - \mu)\,[\nabla_\theta \phi(\theta \mid \mu, \Sigma)]^\top - \phi(\theta \mid \mu, \Sigma)\,\Sigma^{-1} \\ &= \phi(\theta \mid \mu, \Sigma)\,\Sigma^{-1}(\theta - \mu)(\theta - \mu)^\top \Sigma^{-1} - \phi(\theta \mid \mu, \Sigma)\,\Sigma^{-1}\,. \end{aligned} \tag{14}$$

In addition, we can also compute the gradient w.r.t. the parameters $\mu$ and $\Sigma$:

$$\begin{aligned} \nabla_\mu \phi(\theta \mid \mu, \Sigma) &= \phi(\theta \mid \mu, \Sigma)\,\nabla_\mu\big(-\tfrac{1}{2}\theta^\top \Sigma^{-1}\theta + \theta^\top \Sigma^{-1}\mu - \tfrac{1}{2}\mu^\top \Sigma^{-1}\mu\big) \\ &= \phi(\theta \mid \mu, \Sigma)\,\Sigma^{-1}(\theta - \mu) \\ &= -\nabla_\theta \phi(\theta \mid \mu, \Sigma)\,, \end{aligned} \tag{15}$$

and

$$\begin{aligned} \nabla_\Sigma \phi(\theta \mid \mu, \Sigma) &= \frac{1}{\sqrt{(2\pi)^d \det\Sigma}}\nabla_\Sigma \exp\big[-\tfrac{1}{2}(\theta - \mu)^\top \Sigma^{-1}(\theta - \mu)\big] \\ &\quad + \exp\big[-\tfrac{1}{2}(\theta - \mu)^\top \Sigma^{-1}(\theta - \mu)\big]\,\nabla_\Sigma \frac{1}{\sqrt{(2\pi)^d \det\Sigma}} \\ &= \frac{1}{2}\,\phi(\theta \mid \mu, \Sigma)\,\Sigma^{-1}(\theta - \mu)(\theta - \mu)^\top \Sigma^{-1} - \frac{1}{2}\,\phi(\theta \mid \mu, \Sigma)\,\frac{1}{\det\Sigma}\nabla_\Sigma \det\Sigma \\ &= \frac{1}{2}\,\phi(\theta \mid \mu, \Sigma)\,\Sigma^{-1}(\theta - \mu)(\theta - \mu)^\top \Sigma^{-1} - \frac{1}{2}\,\phi(\theta \mid \mu, \Sigma)\,\Sigma^{-1} \\ &= \frac{1}{2}\,\nabla_\theta^2 \phi(\theta \mid \mu, \Sigma)\,. \end{aligned} \tag{16}$$

## B   Experimental details

The numerical experiments conducted in this paper are implemented with PyTorch in Python, using a 2023 MacBook Pro with Apple M2 Pro chip and 32GM memory. The fully-connected layers are implemented using PyTorch's built-in functions. The GM layers are implemented using the derivation in Appendix A (thanks to PyTorch's Automatic Differentiation engine, we only need to implement the loss function, and there is no need to implement the gradients explicitly).

The error bars in Figures 2, 3, 5, 6 and 7 represent one standard error computed over 5 independent trials. When implementing a two-layer GM network, the output of the first GM layer is normalized to have unit norm, before being sent to the second layer as input.

Finally, we include the architecture of the two network structures used in the CIFAR experiments. The VGG-like network structure is as follows:

- **Conv Layer 1:** Input: $32 \times 32 \times 3$ CIFAR-10 image; Convolution: 64 filters, $3 \times 3$ kernel, padding 1; Output: $32 \times 32 \times 64$; Batch Normalization; ReLU Activation.

- **Conv Layer 2:** Input: $32 \times 32 \times 64$; Convolution: 128 filters, $3 \times 3$ kernel, padding 1; Output: $32 \times 32 \times 128$; Batch Normalization; ReLU Activation; Max Pooling: $2 \times 2$; Output: $16 \times 16 \times 128$.

- **Conv Layer 3:** Input: $16 \times 16 \times 128$; Convolution: 256 filters, $3 \times 3$ kernel, padding 1; Output: $16 \times 16 \times 256$; Batch Normalization; ReLU Activation.

- **Conv Layer 4:** Input: $16 \times 16 \times 256$; Convolution: 256 filters, $3 \times 3$ kernel, padding 1; Output: $16 \times 16 \times 256$; Batch Normalization; ReLU Activation; Max Pooling: $2 \times 2$; Output: $8 \times 8 \times 256$.

- **Fully Connected Layer 1:** Input: 16384 (after flattening); Output: 256; ReLU activation; Dropout: 0.5.

- **Fully Connected Layer 2:** Input: 256; Output: 128; ReLU activation.

- **Fully Connected Layer 3:** Input: 128; Output: 10 (for CIFAR-10) or 100 (for CIFAR-100).

To incorporate the GM layer into the above structure, we replace the Fully Connected Layers 2 and 3 with a GM layer (10 components). When training these two networks, we use data augmentation techniques including random horizontal flips and random cropping.

