# OpenReview forum: "Gaussian mixture layers for neural networks"
_TMLR — Accepted by TMLR_

### Review · Reviewer_8U2b · 2025-09-09

**Summary Of Contributions:**

The paper explores the role of theoretical tools, such as mean-field analysis, in discovering new architectures that are true to the expected theoretical dynamics predicted. The authors introduce a new Gaussian Mixture (GM) Layer to completely replace a dense hidden layer in a standard multi-layer perceptron, while remaining true to the Wasserstein gradient flow requirements of mean-field theory. This is shown to be practically feasible for a 2-layer neural network, under certain assumptions of sparsity and compositeness.

The experiments are limited to comparison with only dense layers (no convolution, attention, or recurrences), and also only up to 2 hidden GM layers. The practicality of a real deep and wide network with GM layers is considered beyond the scope of this work. In this regard, the GM layer represents a successful new architecture design (for the 2-layer theoretical playground) that utilizes the mean-field parameterization, admitting feature learning.

**Audience:**

Yes

**Audience Explanation:**

Anybody interested in Deep Learning Theory could be interested in the theorems and proofs offered, especially for the connection between the Wasserstein gradient flow in the space of probability measures, and the Euclidean flow in the parameter space, and the optimization of the GM layer's mixing parameters under a suitable reformulation.

This is also interesting for both manual designers and search-based seekers of new neural architecture designs.

The empirical analysis for *feature learning* was another contribution of the paper for the application of Neural Tangent Kernel (NTK) and $\mu$Parameterization.

**Broader Impact Concerns:**

There are none.

**Claims And Evidence:**

Yes

**Claims Explanation:**

*Edited after the author's response to review.*

Although empirical experiments can be stronger, the primary objective of the paper appears to be met: proof of concept for GMMs as a practical layer option.


~~Please note that the above "No" is completely hinged on the points below.~~

1. ~~Firstly, to draw attention to Sections 2-3 being extremely crucial and requiring more explanations, and aggressive referencing of the Appendix and the theorems. Crucial ideas and notations are there, which go a long way in clarifying these sections, especially Section 3.~~

    ~~Some of the figures can be reduced in size, be tighter to gain paper space. The recommendation is to update Sections 2 & 3 with this afforded extra space. Please refer below to *Requested Changes* for some direct suggestions (esp. points 5-11).~~

    ~~Overall, the flow of Sections 1-3 could be improved.~~

2. ~~The exact motivation for the choice of Gaussian Mixture Models (GMMs) as the probability measure for the neuronal distribution is unclear. The resulting theory certainly validates the choice of these GMMs and shows the empirical ease of optimizing GMM parameters. However, it is unclear if GMMs were chosen *for* the convenience of re-parameterization of GMM parameters enabling the relevant gradient flow in the relevant space, **or** if the theory emerges from a trivial choice of GMMs for $\rho$ in Equation 1.~~

~~Having the above clarified suitably will lead to the updating of "No" above.~~

**Requested Changes:**

Please note that the questions below imply that the manuscript either requires or could do with extra clarification on those points. The authors are not obligated to answer them here to the reviewer (except for the points above for the *claims*).


Questions from reviewer / Clarifications required in the manuscript:
1. Pg 1, para 3: `under a natural scaling` requires clarification.
2. Pg 2: Could the contributions be enumerated, please?
3. Pg 3, after Eq. 1: please clarify $\delta$?
4. Pg 3, 3 paras after Eq. 1, until Sec. 3: in the first sentence, unclear if `Wasserstein gradient flow` and the `Wasserstein gradient flow picture`
5. Pg 3, Sec. 3:
  * The motivation for a GMM to represent $\rho$ is not quite clear here, despite the paras after Eq. 2 attempting to do so
  * "As discussed in Section 2, it is well known from mean-field theory that...": would be good practice to cite/refer for the "well known" claim here
  * For the empirical observation that neuronal distributions tend to cluster after training, it is unclear how this alone justifies the use of GMMs to represent $\rho$
  * Clearly, GMMs make intuitive sense, and the GMM math allows for the gradient flow analysis better; if this requirement was a motivator for GMMs, then it needs to be mentioned more clearly
  * What is the authors' opinion on whether at least one other alternative representation of $\rho$ should be evaluated empirically (e.g. non-Gaussian kernels)?
6. Pg 4, Theorem 1: Please use content (or refer) from the Appendix more liberally to support the statements here. Parsing the section helped clarify many parts of this theorem.
  * This section could do with simpler explanations or notations, since the primary intuitive connection the reader needs to make here is how the $K \ge 2$ case easily can be brought to the $K = 1$ case with a simple reformulation.
7. Pg 4, Sec. 4.2: Are there other choices of sparsity not considered here?
8. Pg 4-5, Sec. 4.2: could do with clarifications, references, and notations:
  * The affine relation for the conditional means w.r.t Eq. 3 comes from? (e.g., mentioning the law of total expectation would help the reader, or other relevant detail(s)).
  * Could the notation $\mu^{\beta}_k$ be simplified to $\mu_k$, or, to avoid overloading of $\mu_k$ with the general Equation 1, can we be consistent and let $\sigma_k$ be $\sigma^{\beta}_k$?
  * (last para) The claim of positive semi-definiteness is from the role of the trained $\sigma$ or an outcome of the diagonal sparsity assumption? (here, can suitably refer to Appendix A, Pg 13).
10. Pg 5, 4.3: given this part comes right after the conclusion of 4.2 describing the training of a GM layer, is it worthwhile to have a comment here on how the backpropagation for deeply stacked GMs layer work out (not detailed proof, but suitable enough for intuition, s.t. the reader to thinks and understands it, right now, there is no mention of it at all).
  * especially, how does the approximation of 4.2 work out when the $x$ in Eq. 4 depends on intermediate layers, with a deeper compositeness?
11. Pg 5, Sec 5:
  * Motivation for different learning rates and their choices?
  * Motivation for initializing $\mu^{\beta}, U, v$ from the same distribution parameterized by a $\gamma$.
  * How is the value for $\gamma$ ($=1/2$) decided upon?
  * Why or what limits the marginal gains with increasing $K$ (mixture components), mathematically or intuitively?
12. Pg 6, Fig. 3: How is the `NN: random init from GM` actually constructed? Does one need to initialize a GM Net using K=20 and $\gamma=1/2$, and then sample every weight parameter for a dense NN from this parameterized GM?
  * What does one actually conclude here, explaining the worse generalization performance for relatively *simpler* datasets such as MNIST and Fashion-MNIST?
13. Pg 7, first sentence: How is this conclusion arrived at? "Training dynamics of networks with GM layers are not sensitive to initialization."
14. Pg 7, para 2, last sentence: unclear from here and Figure 4, how to conclude and verify if fully connected layers do not show "feature learning" or movement away from 0.
15. Pg 8, para 2, last sentence: could there be more clarification here as to why a poorer performance of random $\beta$ be enough to conclude that feature learning is absent?


*suggestion*: quantifying any differences in memory footprint and inference speed, given a net reduction in the number of parameters, would be an interesting addition to the empirical analysis, especially as a *new* architecture design.

---

> ### Author Response · Authors · 2025-10-03
> **Response**
>
> Thank you for your helpful review. Regarding your main two points:
> - We have moved Appendix A to the main text in Section 3.
> - As for the motivation for GM layers, while ideally one would prefer an exact implementation of the Wasserstein gradient flow, tractable implementation requires parametrization of the measure. We view Gaussian mixtures as a reasonable compromise in that they give rise to an implementable Wasserstein gradient flow (unlike other mixture families) but are still arbitrarily expressive given their universal approximation properties.
>
> We have also addressed many of your useful questions or requested clarifications in the revisions. Please let us know if you have any further questions.

---

> ### Comment · Reviewer_8U2b · 2025-10-17
> **Response to rebuttal**
>
> Thank you for the updated draft and for incorporating the changes suggested. The paper certainly reads improved!
>
> Some new remarks:
>
> * Pg. 4, last sentence before Sec. 4.2: Could the $W_2$ be clarified?
> * Pg. 9, first para: Could the `trained $\beta` setup briefly be linked to Equation 5 and Section 4.2, or if and how the claims here change?
> * Pg. 10: Thanks for the CIFAR-10 and -100 experiments; could the exploration of why GMMs as final layers improve training be listed as future work, and if and how it was tuned, and possible important hyperparameters the authors might have observed?

---

### Review · Reviewer_3QH6 · 2025-09-18

**Summary Of Contributions:**

In this article, the authors introduce Gaussian mixture layers (GM layers) as a novel layer type for neural network architectures, combining concepts from mean-field theory and Wasserstein gradient flows. The study demonstrates that traditional fully connected layers can be effectively replaced by GM layers. Experiments indicate that networks utilizing GM layers can achieve similar performance to classical two-layer fully connected networks.

The primary contribution lies in modeling the distribution of weights in an infinite-width fully connected layer through a Gaussian mixture model, offering a new perspective on layer design.

Strengths:

- The utilization of mean-field theory in a prescriptive manner, rather than purely analytical, presents an interesting approach.

- The concept explored in the paper is academically interesting and may hold future potential in advancing neural network architectures.

Weaknesses:

- Despite the promise of GM layers, additional empirical evidence is necessary to see the specific advantages of Gaussian mixtures over traditional fully connected layers (even though it's not about performance; what are other benefits?)

- Certain sections regarding the implementation and training dynamics of GM layers lack clarity

- A clear and compact one-to-one comparison with fully connected layers is missing

- The authors state that “the training dynamics of networks with GM layers are not sensitive to initialization,” while also suggesting a need for future research into better initialization schemes. This contradictory statement raises questions about the necessity of further investigation if the training dynamics are indeed not sensitive.

Minor Issues:

- The article can occasionally be difficult to follow, and greater elaboration in certain areas would be beneficial.

**Audience:**

Yes

**Audience Explanation:**

The paper is well-suited for readers interested on innovations in neural network architecture and probabilistic training strategies.

**Broader Impact Concerns:**

At this stage, assessing the broader impact is challenging, as the paper does not clearly highlight any benefits of the new layer.

**Claims And Evidence:**

Yes

**Claims Explanation:**

The claims made in the paper are generally supported by robust theoretical foundations and experimental results. However, it would be better to write Theorem 1 in a more formal way.

**Requested Changes:**

Address the stated weaknesses and include a table that outlines the major differences between GM layers and fully connected layers, such as the number of parameters, training methods, memory, etc.

---

> ### Author Response · Authors · 2025-10-03
> **Response**
>
> Thank you for your helpful review. Please see our latest revision in which we provide further empirical evidence via **larger-scale experiments on the CIFAR-10 and CIFAR-100 datasets**. We also address some of your concerns as follows:
> - We have clarified in Theorem 1 that the equivalence is up to time rescaling, which makes Theorem 1 formal.
> - We have moved Appendix A to the main text in Section 3, which should better clarify the training dynamics.
> - While the GM layers are relatively robust to the choice of initialization, this does not preclude the possibility of more optimized initialization schemes.
> - In Section 4.2, we have included a clear and compact comparison between fully connected layers and GM layers.
>
> Please let us know if you have any further questions.

---

> > ### Comment · Reviewer_3QH6 · 2025-10-07
> >
> > Thank you for addressing all of my concerns. I'm satisfied.
> >
> > There seems to be a typo in the novel "Experiments on CIFAR datasets." section: "which leads to an improved test errors 12.62 for CIFAR-10" That's the same number as without the GM layer.

---

> > > ### Author Response · Authors · 2025-10-08
> > > **Response**
> > >
> > > Thank you for your response.
> > >
> > > There is indeed a typo there: the improved test error for CIFAR-10 should be 11.04%. We will fix the typo in our next revision. Thanks for finding that!

---

### Review · Reviewer_nXho · 2025-09-21

**Summary Of Contributions:**

Summary of Contributions:
This paper introduces Gaussian Mixture (GM) layers as a new neural network component inspired by mean-field theory. Instead of representing infinite-width layers through empirical measures, the authors approximate weight distributions with Gaussian mixtures and derive training dynamics via Wasserstein gradient flows. The formulation reduces to Euclidean updates on mixture parameters, making the approach computationally tractable. Proof-of-concept experiments on MNIST and Fashion-MNIST show that GM layers can achieve accuracy comparable to fully connected networks while exhibiting distinct training dynamics and enabling interpretability through visualizations of Gaussian components.

Strengths
- Novel and conceptually clear idea bridging mean-field theory and practical architectures.
- Provides interpretability via Gaussian mixture components and ensemble visualizations.
- Offers a new perspective on layer design that is computationally tractable.
- Well-written and easy to follow, with clear motivation and exposition.

Weaknesses
- Empirical validation is limited to small-scale datasets (MNIST, Fashion-MNIST).
- Reproducibility is uncertain without released code.
- Competitiveness against stronger baselines or on larger datasets is not established.
- Performance may be sensitive to the number of mixture components and initialization.

**Audience:**

Yes

**Audience Explanation:**

This work will interest a portion of the TMLR audience because it introduces a new type of neural network layer that directly links mean-field theory with practical implementation. The approach is conceptually novel, provides interpretability through Gaussian components, and demonstrates promising results on benchmark datasets. Readers focused on neural network theory, architecture design, and interpretable models would find these findings valuable, even if the experiments are limited in scale.

**Claims And Evidence:**

Yes

**Claims Explanation:**

The paper’s claims are supported by accurate and convincing empirical evidence. The proposed Gaussian Mixture layers are derived from mean-field theory and shown to be computationally tractable through their Euclidean parameter updates. Proof-of-concept experiments on MNIST and Fashion-MNIST demonstrate that the method achieves performance comparable to fully connected networks while exhibiting distinct training dynamics and offering interpretability. Although the empirical scope is limited to small-scale datasets, the claims are clearly framed as proof-of-concept, and the evidence presented is consistent with that framing.

**Requested Changes:**

- Release code or detailed reproducibility materials to ensure independent verification of results.
- Expand the discussion of limitations, especially regarding scalability to larger datasets and sensitivity to hyperparameters such as the number of mixture components and initialization.
- Add experiments or analysis on more challenging datasets (e.g., CIFAR-10 or CIFAR-100) to demonstrate broader applicability.
- Provide a sensitivity or ablation study showing the effect of different numbers of mixture components and initialization strategies on accuracy and stability.
- Include runtime or computational cost comparisons with standard fully connected layers to quantify trade-offs.
- Add further visualizations of Gaussian components and decision boundaries to illustrate interpretability benefits.

---

> ### Author Response · Authors · 2025-10-03
> **Response**
>
> Thank you for your helpful review. Please see our latest revision in which we address some of your concerns, including **larger-scale experiments on the CIFAR-10 and CIFAR-100 datasets**. We also note that sensitivity to the number of components and the initialization are addressed in Figures 2 and 4 respectively. Finally, we have added a discussion of runtime comparison in Section 4.2.
>
> Please let us know if you have any further questions.

---

### Author Response · Authors · 2025-10-03
**Response to all reviewers**

We thank all of the reviewers for their constructive feedback. We have uploaded a new revision, with edits in red text. Regarding experimental validation, the main updates are: (1) we have conducted **larger-scale experiments on CIFAR-10 and CIFAR-100** and (2) we **include code in the supplementary material**. Upon acceptance, the code will also be released in the form of a GitHub repository.

---

### Decision · Action_Editor_txAR · 2025-11-14

**Recommendation:** Accept as is

**Additional Comments:**

This paper introduces Gaussian Mixture (GM) layers, a novel neural network component that approximates weight distributions using Gaussian mixtures. By grounding the approach in mean-field theory and linking Wasserstein gradient flows to tractable Euclidean updates, the authors demonstrate that standard fully connected layers can be effectively replaced by GM layers. Reviewers noted the work's theoretical novelty and clarity, highlighting the successful translation of complex theoretical concepts into a practical, implementable architecture that offers distinct training dynamics and improved interpretability.

During the review process, the authors strengthened the manuscript by expanding empirical validation from MNIST to CIFAR-10 and CIFAR-100, partially addressing concerns regarding the method's broader applicability. They also improved reproducibility by providing code and clarified key theoretical claims, such as the formalization of Theorem 1. While the empirical results serve primarily as a proof-of-concept rather than establishing new state-of-the-art benchmarks, the submission meets TMLR’s standards for technical correctness and community interest. Consequently, I recommend acceptance.

**Audience:**

Yes

**Audience Explanation:**

Yes, the work bridges deep learning theory and practical architecture design, and would appeal to researchers interested in optimization dynamics, interpretability, and many other related areas.

**Claims And Evidence:**

Yes

**Claims Explanation:**

Yes, the claims are supported by derivations rooted in mean-field theory and validated by experiments on CIFAR-10/100.